# Design of a Pyroacuotubular (Mixed) Boiler for the Reduction of Flue Gas Emissions through the Simultaneous Generation of Hot Water and Water Steam

**Duilio Aguilar Vizcarra** [1], **Doris Esenarro** [2] and **Ciro Rodriguez** [3,*]

1 Faculty of Mechanical Engineering, Universidad Nacional de Ingeniería UNI, Rimac 15333, Peru
2 Facultad de Arquitectura y Urbanismo, Universidad Ricardo Palma—URP, Santiago de Surco 15039, Peru
3 Department of Software Engineering, Universidad Nacional Mayor de San Marcos UNMSM, Lima 15081, Peru
* Correspondence: crodriguezro@unmsm.edu.pe

**Abstract:** Environmental protection is a continuous challenge that requires innovating the combustion process of boilers that emit polluting gases. This research proposes a novel pyroacuotubular (mixed) boiler design that reduces the emission of combustion gases by hot water and steam. The applied methodology considers the dimensioning-construction, modification, and analytical calculation of water volume, metallic masses, heat for hot water and steam generation, and combustion gases. The Ganapaty method of heat transfer is applied to prioritize the velocity of gas displacement, the pressure drop along the pipe, and its application on surfaces. In the parallel generation of hot water and steam, a mass of $CO_2$ (1782.72 kg/h) and CO (5.48 kg/h) was obtained; these masses were compared with the results of the proposed design, obtaining a reduction in the mass of gases emitted to the environment in hot water $CO_2$ (44.35%) and CO (44.27%); steam $CO_2$ (55.65%) and CO (55.66%). A significant reduction was achieved in the simultaneous generation of hot water and steam compared to the individual generation, which shows that the simultaneous generation of the pyroacuotubular (mixed) boiler reduces the emission of combustion gases.

**Keywords:** pyroacuotubular boiler; combustion gases; pollutant gases; $CO_2$; CO; environmental pollution

## 1. Introduction

Environmental pollution caused by the emission of greenhouse gases, including combustion gases and their temperature, has a negative impact on the environment, generating environmental pollution and contributing to the greenhouse effect due to the presence of carbon dioxide. Generally, this pollution is generated by industrial processes that use hydrocarbons, for example, boilers, which increase the AQI (Air Quality Index), and are considered an index between 75 and 100 for a polluted environment [1].

Among the most common sources of pollutants emitted into the environment, we can mention combustion engines of various types, which are widely used in industry, such as the pharmaceutical industry, agro-industry, textile, fishing, tire manufacturers, oil companies, thermal power generation plants, pointing out in these cases, the effects they can cause to the environment [2]. In addition, different air quality control plans applied in other countries are mentioned in this work, pointing out the equipment that should be considered to eliminate the negative aspects caused by the quality of water, atmosphere, and soil. Cities and populations require ecological and sustainability, which are important in the mitigation of $CO_2$ caused by polluting gases, and one of the generators are boilers [3].

Greenhouse gas emissions and methane emission rates present specific and regionalized reduction opportunities, which provide greater resolution by focusing on specific

geographic units [4]. The behavior of carbon dioxide is as a gas in air at standard temperature and pressure or as a solid (dry ice); if the temperature and pressure are increased from the standard point to the critical point of carbon dioxide, it can adopt properties between a gas and a liquid and behave as a supercritical fluid above its critical temperature, in this way supercritical carbon dioxide ($sCO_2$) maintains its critical or above standard temperature and pressure. Supercritical carbon dioxide ($sCO_2$) uses $CO_2$ as the working fluid in small turbomachines. Power cycles based on sCO2 have the potential for increased heat-to-electricity conversion efficiency, high power density, and simplicity of operation compared to existing steam-based power cycles. In turn, Thanganadar [5] analyzes the peak performance and cost of electricity from five cascaded cycles. The pressure ratio of the Gas turbine GT is optimal to obtain the cost reduction in the $sCO_2$ cycle for the equivalent steam. The new $sCO_2$ cycle configuration provides ideal temperature slip in the bottom cycle heat exchangers and cycle efficiency. In the research of [6], the methods analyzed in $CO_2$ and the principles of these methods are considered for application in new low-emission energy technologies.

$CO_2$ emissions also arise from some industrial processes, as shown in Figure 1, resource extraction, and burning forests during land clearing. Some sources could supply decarbonized fuel, such as hydrogen, to the transportation, industrial, and construction sectors, thereby reducing emissions from these distributed sources. Some industrial processes can also use and store small amounts of $CO_2$ captured in manufactured products. $CO_2$ is also emitted during certain industrial processes, such as cement manufacturing, hydrogen production, and biomass combustion. Power plants and other large-scale industrial processes are prime candidates for $CO_2$ capture and storage.

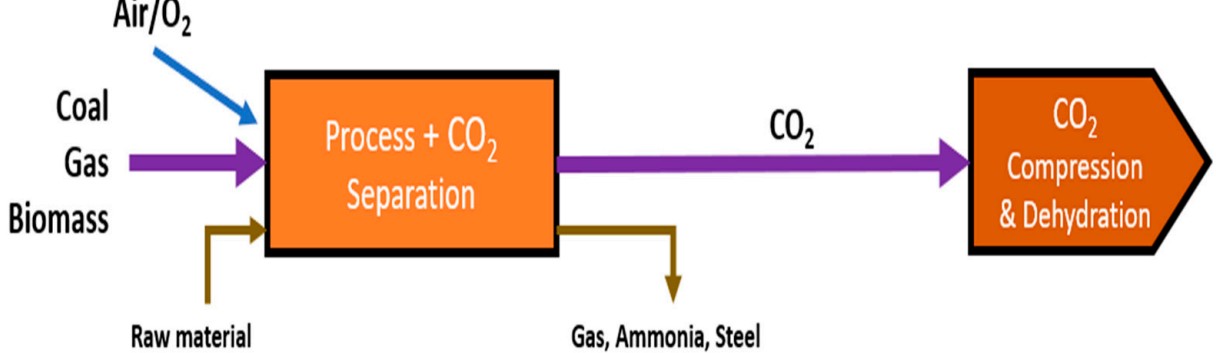

**Figure 1.** $CO_2$ capture systems in industrial processes (adapted from BP).

There are several techniques to reduce $CO_2$ pollution; among the most developed are post-combustion and oxy-combustion, pre-combustion and industrial processes from all compressing and dehydrating. For NOx reduction, the CSR (Selective Catalytic Reduction) method can be used, which has an efficiency between 90 and 98%, also combined with NSR (Nox Storage Reduction) catalysts.

Carbon Capture and Storage (CCS) is an alternative industry strategy that can deeply cut industrial $CO_2$ emissions, and CCS is moving into carbon dioxide removal (CDR) in applications such as Direct Air Capture (DAC) and Bioenergy with CCS (BECCS), drawing down historical CO2 emissions from the atmosphere.

Oxyfuel Combustion (Figure 2) is another alternative to convert fossil fuel into carbon-neutral or carbon-negative with $CO_2$ utilization/storage, mitigating the climate change issue and contributing to the environment and economy of the community.

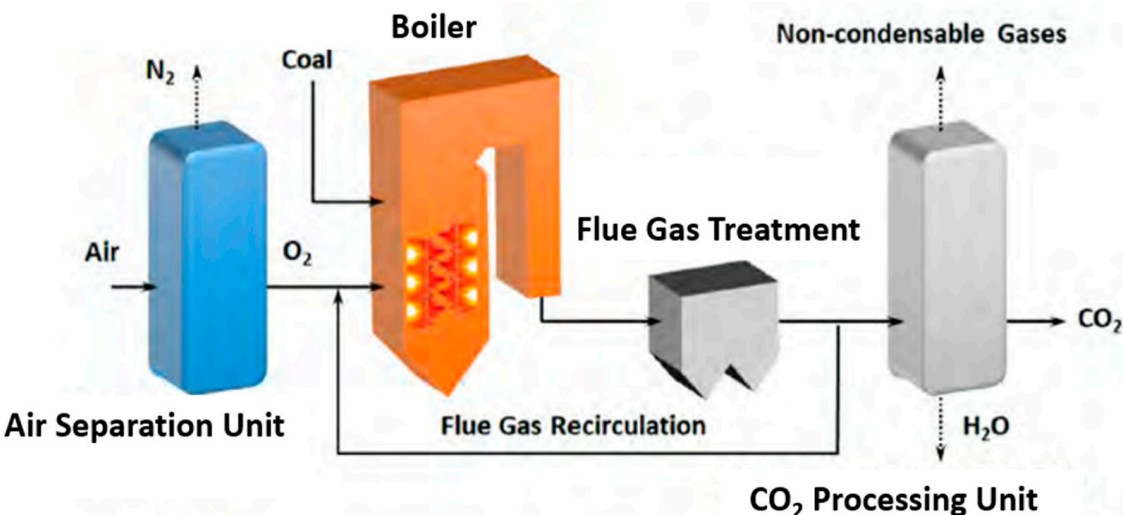

**Figure 2.** Oxyfuel combustion technology. Source: Technical Report of State of the Art: CCS Technologies 2022. Global CCS Institute.

On the other hand, we can identify that $CO_2$ is a critical factor in climate change and it is promoted by fossil fuels; forestry systems can absorb carbon, and biomass is an interesting alternative to mitigate the problem of climate change [7]. An equivalence of 718 kg $CO_2$ per person per year was estimated in the city of Ibagué, Colombia, based on fuel service stations for each type of fuel (gasoline, diesel, and natural gas), which are generated by small vehicles [8].

There is a growing development of technology dedicated to controlling pollutant gas emissions. This technology imposes electronic systems that help control greenhouse gas emissions. These electronic control devices implement vehicles and equipment such as boilers [9]. It is also mentioned that this electronic system helps the automotive industry reduce the emission of polluting gases; in the future, the use of vehicles known as hybrids to help the specialized technician or user determine that these gases do not exceed the established standards [10].

In Peru, it has been observed that the industrial manufacturing sector of medium and small companies requires sophisticated equipment that is efficient and contributes to sustainability. Due to their limited economic capacities, this is not always possible [11]. On the other hand, sustainability in this sector is the product of continuous research, which, in the case of industrial boilers, is little explored, observing equipment that generates a large emission of gases into the environment, high fuel consumption, oversizing of boilers, additional equipment to generate hot water or steam, among other aspects.

In the article on solar collectors, the author identifies that steam generation is part of the industrial processes, seeking to improve the efficiency in obtaining steam. To this end, this work points out the use of complementary systems to those used for a steam generation that does not consume fuels [12]. The proposal established mentions the use of solar collectors to increase the temperature of the water feeding the steam generator. In addition, a feasibility analysis was carried out, which showed the high cost of implementing these complementary systems. However, according to this work, they must demonstrate a sustainable approach; otherwise, additional equipment will be necessary to generate this steam [13].

Considering the additional equipment needed by industries with boilers to generate hot water or steam, these cause losses in production time, additional labor costs, higher maintenance costs, and investment costs. Moreover, due to deficiencies in their respective processes, these types of equipment generate polluting gas emissions, contributing to environmental pollution [14]. Likewise, it generates an additional cost that, often, the industrial sectors cannot sustain. Because of the problems observed in this sector, it is

mentioned that innovative designs and environmental policies are necessary for better regulation of environmental pollution [1,15].

Strategic Planning that considers the territorial approach of a country such as Peru to access OECD focuses on the results that require the country. The Strategic Plan for National Development, among others, includes the sustainable use of biological diversity and a healthy environment and water resource management, it considers that environmental and sustainable policies should be a priority, and considers that consolidating the productive economic system should be used sustainably alongside external resources, within a sustained economic growth [16].

The simultaneous generation of hot water and steam can be included in the concept of cogeneration, in which internal combustion equipment is used to generate steam and hot water [17]. This concept is based on the Carnot cycle (Figure 3), where only work is produced, and the waste heat is emitted to the environment in a conventional internal combustion engine. It also illustrates an internal combustion machine where not only work but also heat is produced in the cogeneration process [18].

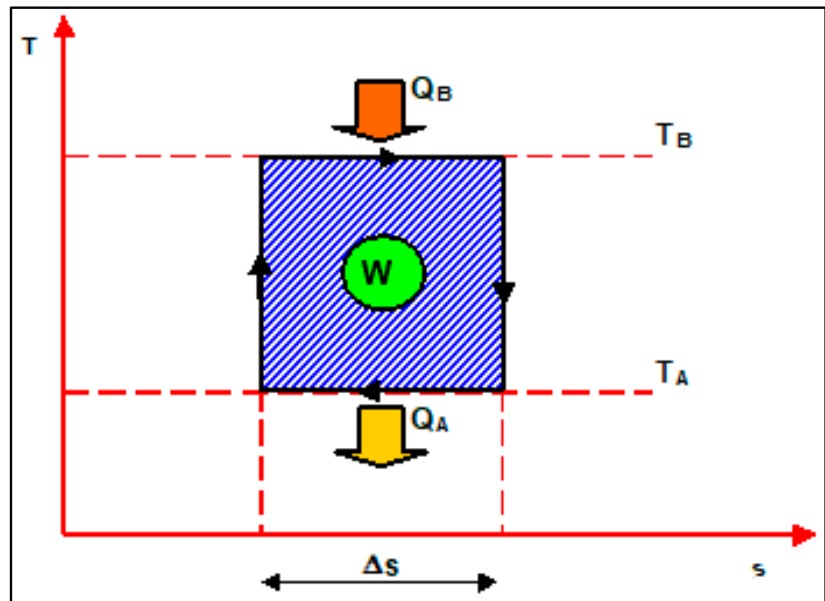

**Figure 3.** Carnot cycle shows the transformation of a higher temperature heat source into work in the shaft of a thermal machine and the rejection of the waste heat to the environment (Adapted from [7]).

It is also necessary to have a heat recovery system that will provide steam and hot water for various uses in an applied production center.

Solar boilers designed for steam generation are used in power plants that use renewable energies and thus eliminate the emission of combustion gases that affect the environment due to the combustion of hydrocarbons [19–22]. However, the company mentions that within the concept of renewable energies, it is possible that the use of Aero generators of the paddle type does not cover the power needs required for the generation of electric energy and given the importance of covering this type of demand and the need to reduce the emission of gases that increase the greenhouse effect, a new type of boiler design that eliminates the emission of combustion gases has been proposed [23].

This power generation plant [24], from the absorption of solar energy and its storage in special salt tanks in a cold tank and a hot tank, can generate steam to be used in the drive of a steam turbine that transforms the energy of steam pressure into energy in the shaft of the machine that is connected to an alternator can generate electricity, as can be seen in Figure 4 [25].

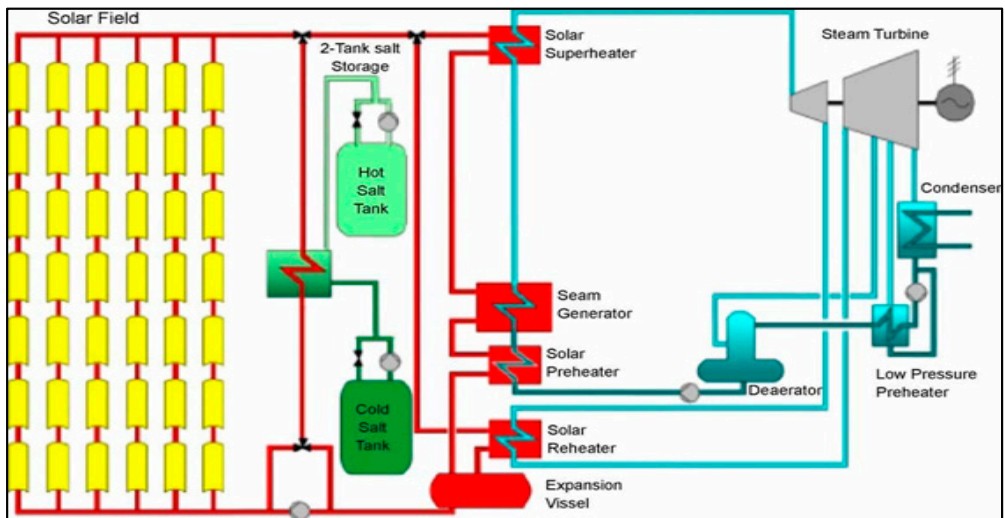

**Figure 4.** Schematic of a plant for generating electricity from solar energy using steam to drive a steam turbine with a solar heat source, Source: (Herrmann, 2002).

In the present investigation, taking as a reference the gas data obtained in the experiment of [1], we seek to improve its design since, in the previous work, hot water and steam were generated alternatively, while in the design proposed in the present investigation, hot water, water steam, and thermal oil are generated simultaneously [6].

In order to answer the research question about how to improve the thermal efficiency of a boiler by reducing the emission of combustion gases in the generation of hot water and steam?

To achieve the goals of the research, specific objectives were defined that will drive the research and help to narrow the focus and guide the research process; in that sense, the following specific objectives have been set:

1. Simultaneously generate hot water and water steam; this objective is achieved with a new design of the three-pass pyroacuotubular (mixed) boiler with LPG fuel that establishes the heat needs required by the metallic mass and water to generate hot water and steam simultaneously based on the Zero Law of Thermodynamics to reach thermal equilibrium.
2. Reduce the temperature of the gases emitted by increasing the thermal efficiency of the pyroacuotubular (mixed) boiler; better use of the combustion gases is achieved by generating hot water and steam at a higher speed reducing the flue gas emission temperature of the flue gas.
3. Reduce the emission of greenhouse gas pollutants; the reduction in the gas temperature generates a better use of the combustion gases and, at the same time, a lower environmental impact, compared to other industrial boilers, contributing to reducing global warming and environmental pollution.

## 2. Materials and Methods

The applied methodology has, as its starting point, the dimensioning construction (mixed boiler), modification (combustion chamber), and analytical calculation (water volume, metallic masses, heat for hot water and steam generation, combustion gases) [6].

### 2.1. Instruments

The instruments used were documented and described, detailing technical aspects.

- Pressure gauge, Bourdon type, range 0–200 psi (0–1.378 MPa), with an approximation of 0.01378 MPa;
- Thermometer, Bimetallic type, range 0–150 °C, approximating 2 °C;
- Balance, Mirway type, range 0–130 kg, model BM105B, and 1kg approximation;

- Gas analyzer calculates $\%CO_2$, $\%CO$ and $\%O_2$;
- Chronometer;
- LPG fuel.

In addition, to validate the experimental results, a fuel gas evaluation analytic model was developed.

### 2.2. Procedures

The procedure evaluates the stages of dimensioning construction, modification, analytical calculation, and comparative evaluation, as shown below in Figure 5.

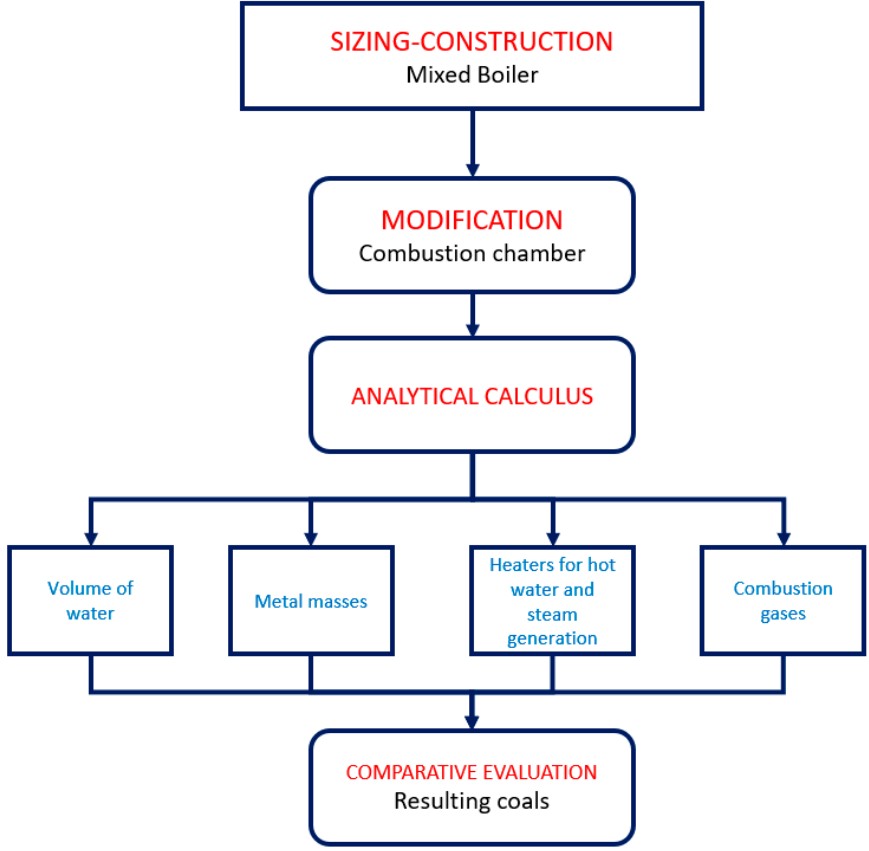

**Figure 5.** Processes considered.

### 2.3. Boiler Designs

To develop the design of a vertical boiler called a pyroacuotubular (mixed) boiler, which is a combination of a pyrotubular boiler that allows the combustion gases to pass through the interior of the tubes of a pyrotubular boiler to generate steam and an aquotubular boiler because it allows the water to pass through the interior of the tubes in the heating process to become steam [1,24]. Their characteristics are mentioned below:

The water-tube boiler results from the constructive combination of two heat exchangers, where the fluids circulate: one in parallel flow and the second in counterflow. The aforementioned is an improvement in the state of the art as it presents differences from the point of view of design, construction, and operation, to be of simultaneous generation, because it can produce hot water and steam in parallel, with this option reduces the equipment that a user may need of both heat-carrying fluids [25].

The vertical boiler's internal assembly is made up of three fundamental components: a combustion chamber, a water chamber, and a dome. The latter has a protective cover at the top; all these components are removable, facilitating maintenance and repair activities due to the less time spent in disassembling and assembling its parts. Externally, it is

protected with thermal insulation, which is covered with a thin metallic plate called a metallic lining [26].

By assimilating the wide path of the combustion gases, the transfer area achieves better utilization of the combustion heat transferred by the gases, which influences a higher heating rate of the water, where fuel consumption and better utilization of the heat of the combustion gases. Furthermore, the higher temperature resulting from combustion is used in the dome (inside which steam is generated) to increase the speed of steam generation, which results in higher thermal efficiency [27].

Because the combustion gases travel through the boiler's interior, higher thermal efficiency is achieved because the combustion heat is used radially, which results in a double use of the combustion heat. In addition, better combustion is achieved so that the exhaust gases, as a consequence of the combustion, can reduce to the minimum the pollution indexes and lower the temperature of the environment.

The combustion chamber, in its interior, contains an ignition chamber fire. On its external surface will be surrounded by water that will circulate to heat up; this configuration allows it to generate hot water and steam simultaneously. Likewise, this combustion chamber in its upper part is provided with a canalized system with an appropriate geometry that allows the unburned elements (elements that do not combust or residues of an inadequate homogenization) resulting from combustion to be trapped and consequently avoids their discharge into the environment [2,28].

It is a vertical 03-pass boiler because the combustion gases undergo three longitudinal re-circulations in a vertical position through the whole heat transfer area; previously, this area has been subdivided into three parts, and when delivering its heat to these three parts of the transfer area it undergoes three passes (by the ASME Code) of gases at a temperature of 180° to accomplish the passes, to exit through the circular section of the chimney, finally. The use of fuel can be an alternative, i.e., it can burn diesel fuel, liquefied petroleum gas, and natural gas.

### 2.4. Boiler Sizing

To carry out the sizing of the boiler, the size of the unit to be calculated or the power to work is previously established, for which it is necessary to consider as initial information the data obtained in the experimental measurements carried out in the work of [1], these experimental data present the same conditions of fuel flow consumed, which is closely related to the boiler power, and directly related to the heat transfer area, and the steam flow to be generated in standard conditions (100 °C and 1 atmosphere of pressure at sea level).

In general, boilers are fundamentally sized based on the heat transfer area; this implies the calculation of the amount of sheet metal to be used in the construction of the boiler for heat transfer without considering those sections of the boiler that do not fulfill this function. Applying the standardized equivalences established by the ABMA (American Boiler Manufacturing Association), the following is obtained:

1BHP (Boiler Horsepower) = 5 feet$^2$ of the transfer surface, as steam flow 1 BHP (9.8 kW) = 34.5 lb of steam/h or 16.68 kg of steam/h. The design pressure at which the sheet metal thicknesses will be calculated is at the discretion of the designer or steam temperature requirements. In addition, the number of gas passages and the position of the boiler, as well as the type of fuel to be combusted.

Aguilar [1,29] analyzed two theories related to the method of calculation and sizing of the transfer area. In both cases, the option of the construction of pyrotubular and aquotubular boilers is considered, which are mentioned below:

The one established by ASME and ABMA (1960) references the development of the science of materials of 10 feet$^2$ since 1930 and before until nowadays in 5.0 feet$^2$, assumed by the manufacturers of boilers. Ganapaty developed a method based on heat transfer by prioritizing the velocity of gas displacement, the pressure drop along the tube, and its application on finned or finless surfaces.

For the present work, the theoretical option of the ASME Code and the ABMA's stipulations are assumed since the unit proposed for sizing is a pyroacuotubular (mixed) boiler that combines the pyrotubular and aquotubular options, for which it is necessary to establish the required parameters and characteristics to size a boiler that will allow demonstrating the use of combustion heat with two fluids simultaneously capable of transporting heat, with the following technical specifications, as is showed in Table 1.

**Table 1.** Technical specifications.

| Design | Pyroacuotubular (Mixed) |
| --- | --- |
| Power | 15 BHP = 150 kW |
| Transfer area | 75 feet$^2$ = 6.96 m$^2$ |
| Theoretical steam flow | 517.5 lb/h = 234.73 kg/h |
| Design pressure | 110 psig = 0.758 MPa |
| Maximum working pressure | 100 psig = 0.689 MPa |
| Position | Vertical |
| Number of gas passages | 03 |
| Fuels to be used | Liquefied petroleum gas (LPG) |
| Simultaneous thermal fluids | Steam and hot water |

### 2.4.1. Combustion Chamber Sizing

The dimensions of the combustion chamber are established based on two criteria:

First, the aesthetic aspect and uniformity of its dimensions that allow an adequate assembly with the flange of the water chamber must be considered for this purpose; the outer diameter of the water chamber has been considered. Then, the external height will also be calculated according to the needs of the burner inlet and the coincidence in diameter at the outlet with the firebox inside the water chamber [30].

Second, this criterion is given from the circulation of combustion gases; to consolidate the path of the three steps, for this purpose, it will be necessary to establish a circular channel at the top that allows the deposit of solid particles or unburned products of poor combustion. Third, an internal consideration to consider is, above all, the development of the ignition chamber or hearth whose volumetric geometry is complex (deformed L), which is the place where combustion starts and where the radiant flame delivers its heat to the water surrounding the ignition chamber with an independent inlet and outlet of hot water that can be used for different purposes [31]. Finally, for greater accuracy, a 3D calculation is made using a CAD computer program for greater precision, as can be seen in Figure 6.

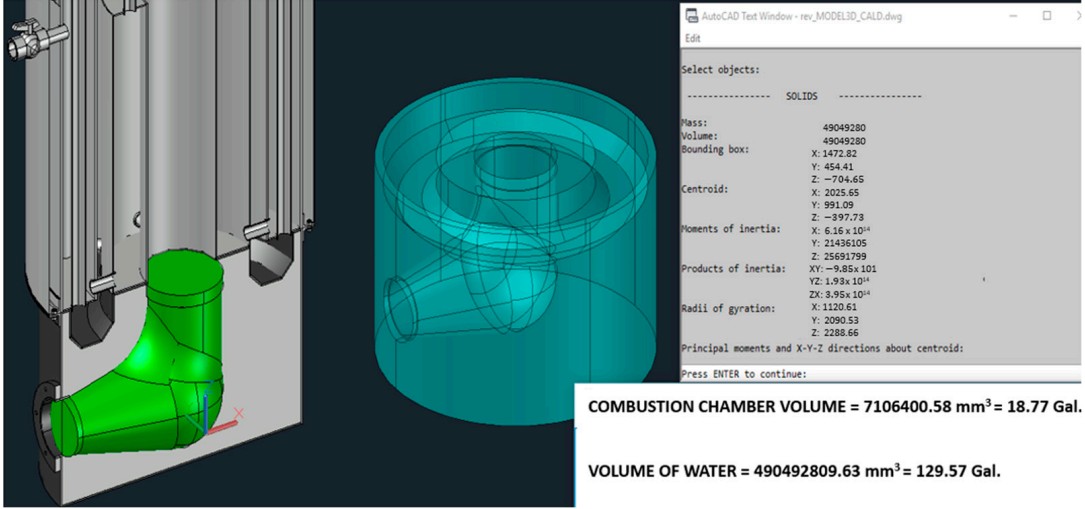

**Figure 6.** Firebox inside the 3D combustion chamber.

### 2.4.2. Water Chamber Sizing

The thicknesses of the metallic plates that constitute the heat transfer area within their cylindrical volumes contain the water in the heating process, and the path of the combustion gas circuit to configure the three gas passages is carried out.

Radius differences are established in the vertical cylinders, as explained in the previous section corresponding to the design. Furthermore, the criteria of the researcher Ganapaty related to the speed of the gases is assumed, which is to reduce the turbulence, reduce the Reynolds number, and assimilate the speed in the range of a laminar fluid; this will allow a slow displacement of the combustion gases and will benefit in greater delivery of heat to the circulating fluid in the heating process and being the gases in contact with two cylindrical surfaces containing water, a radial delivery of heat in two opposite directions is achieved which means a double heat gain.

The following is the calculation and selection of the materials of the three components that make up the fundamental part of the pyroacuotubular (mixed) boiler.

### 2.4.3. Calculation of Plate Thickness and Selection of the Material for the Circular Cylinders

According to the boiler scheme, the plate thicknesses were calculated, applying the ASME code mentioned by Eugene F. Megyes and the formulas for the corresponding calculation. The sum of the cylindrical surfaces, including the tubes that make up the third pass, yields a value of 6.96 m$^2$.

An initial objective is to determine the diameter of the continuation of the hearth, whose length constitutes the first pass of gases; for that, we have as a reference the following: "the hearth is a tube whose diameter should be between 40% and 50% of the diameter of the mirror; the position of this tube or mirror depends exclusively on the design, i.e., it can be lowered or raised along the vertical axis " theory formulated in the Microsoft Encyclopedia Encarta 05 © 2022—2005 of Microsoft Corporation, the design description of "Boiler" was taken and determined that the optimum diameter of the continuation of the ignition chamber is 12 inches, ensuring adequate heat transfer to both outer faces, which configures the aquotubular area corresponding to the boiler [1,32].

The calculation procedure will be executed by the sections and by the location of the cylindrical and other thicknesses, as shown in the following scheme; the values to be applied in the case of vessels subjected to internal pressure are calculated with the following expression, according to the initial parameters:

For ease of understanding and the convenience of comprehension, we have considered the basic and derived SI units, which are independent of each other, as shown in Table 2, and the description of the variables used in the calculations is in Table 3.

**Table 2.** Description of the SI units.

| Quantity | SI Abbreviations |
|---|---|
| Force | N, mm |
| Mass | kg |
| Time | s, h |
| Stress, pressure | Pa, MPa |
| Energy, work, quantity of heat | J, kJ |
| Density | kg/m$^3$ |
| Power | kW |
| Temperature | °C |

**Table 3.** Description of the variables used in the calculations.

| Variable | Description |
| --- | --- |
| Do | Outside diameter |
| t | Plate thickness |
| L | Length of the ignition chamber in the water chamber zone |
| A | Calculation factor (Dimensionless) |
| R | Cylinder radius (variable) |
| t | The thickness of plate. |
| P | Boiler internal pressure |
| Pe | External pressure |
| B | Coefficient resulting from joining factor A and heating temperature. |
| e1, ... e7 | Plate thickness |
| S | Material yield stress, allowable material stress |
| E | Efficiency of the welded joint |
| P | Internal pressure supported by the vessel (0.758 MPa). |
| T | Dome Plate Thickness |
| D | Dome diameter |
| $M_{total}$ | Water mass combustion chamber, water chamber, dome, metallic mass |
| CVF | Calorific value of fuel |
| $\dot{Q}_{theoretical}$ | Theoretical heat required |
| $\eta_{cc}$ | Combustion efficiency |
| $\dot{Q}$, $Q_{total\ real}$ | Total heat |
| $\dot{m}_c$ | Flow Fuel mass |
| $\dot{Q}_A$ | Heat required for phase change |
| $Q_i$ | Heat needed to heat the new volume of water in the dome |
| $\dot{Q}_{sub\ total}$ | Heat required to heat water |
| $\dot{Q}_{total}$ | Total heat required to heat water and generate steam |
| $V_i$ | Internal volume of the steam chamber |
| $\dot{m}_{comb\ real}$ | Actual fuel mass |
| $\eta_{combustion}$ | Combustion efficiency |
| $\dot{Q}_A$ | Heat delivered to generate steam |
| $r_{a/c\ real}$ | Actual air-fuel ratio |
| $r_{a/teorico}$ | Theoretical air-fuel ratio |
| %e | Percentage of excess air |

Design pressure (P) 110 psi = 0.758 MPa; Yield stress (S) = 13,800 psi = 93.8 MPa for ASTM 285 Grade C Carbon Steel; Weld Joint Efficiency = 0.85 (for unexamined joints); R = Cylinder radius (variable); t = Thickness of plate; The formula for internally pressurized vessels is: (Equation (1)); Design pressure (P) 110 psi = 0.758 MPa; Yield stress (S) = 13,800 psi = 93.8 MPa for a carbon steel ASTM 285 grade C; Welded joint efficiency = 0.85 (for unexamined joints); R = Cylinder radius (variable); t = Plate thickness.

The plate thickness of vessels under internal pressure is calculated in Equation (1)

$$t = P \times \frac{R}{SE} - 0.6\,P \tag{1}$$

The vessels subjected to external pressure (Pe) is calculated in Equation (2)

$$Pe = 4 \times B/3(Do/t) \tag{2}$$

where:

　　B = Coefficient
　　Do = Outside Diameter
　　t = Plate Thickness
　　L = Cylindrical part length.

　　In the primary scheme of the boiler, the arrangement of the cylinders on which the thicknesses will be calculated is shown in Figure 7:

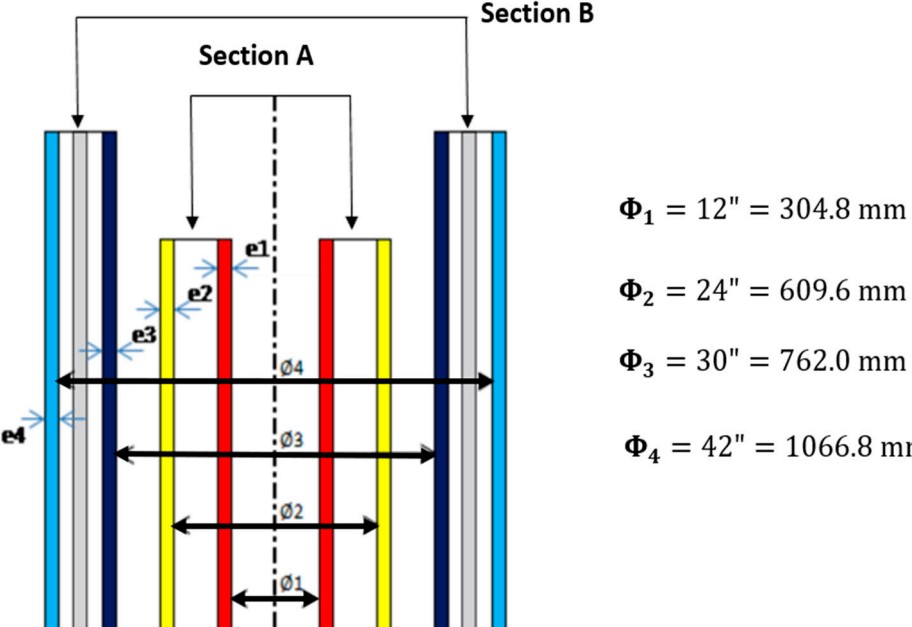

**Figure 7.** Longitudinal section of the cylindrical surface of the vertical boiler, showing dimensions of diameters.

The diameters have been selected to minimize fluid–wall friction losses and consider the water storage volume. The Ganapaty criterion has also been used, which mentions that the pipe diameters in pyro-tubular boilers can vary from 1.5 to 2.5 in. Due to the similarity to these values, the diameters of the cylinders are considered to reduce the friction losses of the combustion gases. The following is the procedure for calculating the thickness of the plates to be used for the circular and elliptical sections.

### 2.4.4. Thickness Calculation (e1)

According to the ASME code for vessels subjected to external pressure and using Equation (3).

$$Pe = \frac{4 \times B}{3(Do/t)} \tag{3}$$

Do = 12″ = 304.8 mm
t = $\frac{1}{4}$″ = 6.35 mm (assumed) will check if it withstands the defined pressure.
L = 40″ = 1016 mm.
We previously calculated: L/Do = 1016/304.8 = 3.33
Do/t = 304.8/6.35 = 48
The value of A = 0.0015
With the value of A and taking a fluid temperature up to 482.22 °C, and the value of B.
B = 6600
Finally, is calculated $Pe = \frac{4 \times 6600}{3 \times 48} = 183.3$ psi = 1.26 MPa

### 2.4.5. Thickness Calculation (e2)

Subjected to internal pressure and using Equation (4)

$$t = \frac{P \times R}{SE - 0.6P} \tag{4}$$

$$t = \frac{0.758 \times 304.8}{93.8 \times 0.85 - 0.6 \times 0.758} = \frac{231.03}{79.28} = 2.91 \text{ mm}$$

S = Allowable material stress (93.8 MPa)
E = Efficiency of the welded joint (0.85 for unexamined joints)

R = Cylinder radius (12″ = 304.8 mm).
P = Internal pressure supported by the vessel (0.758 MPa).

### 2.4.6. Thickness Calculation (e3)

Subjected to external pressure, using Equation (5)
Using the ASME code according to the formula

$$Pe = 4 \times B/3(Do/t) \tag{5}$$

B = Coefficient
Do = Outside diameter (30 inches = 762 mm)
t = Plate thickness assumed (5/16 inches = 7.93 mm)
Also:
L = cylinder length (46 inches = 1168.4 mm)
The ratio length/external diameter and external diameter/thickness of the plate is calculated, respectively.

$$L/Do = 1168.4/762 = 1.53 \; Do/t = 762/7.93 = 96.0$$

With the values calculated, we found the value of
A = 0.0009
With the value of A, considering the temperature up to 148.88 °C is obtain the value of B = 12,200 psi (84.11 MPa)
Finally, it is calculated in Equation (6):

$$Pe = \frac{4 \times B}{3(Do/t)} = \frac{4 \times 12,200}{3(762/7.93)} = \frac{48,800}{288.27} = 1.17 \text{ MPa} \tag{6}$$

### 2.4.7. Thickness Calculation (e4)

Subjected to internal pressure and using Equation (7)

$$t = \frac{P \times R}{SE - 0.6P} = \frac{1 \times 533.4}{93.8 * 0.85 - 0.6 * 0.758} = \frac{533.4}{65.06} = 5.10 \text{ mm} \tag{7}$$

A corrosion factor of 1.25 mm is considered as an additional 1.25 mm to achieve the equivalent of a sheet of $\frac{1}{4}$ inch or 6.25 mm
S = Allowable material stress (93.8 MPa)
E = Welded joint efficiency (0.85 for the unexamined board)
R = Cylinder radius (5334 mm)
P = Internal pressure supported by the vessel (0.758 MPa)

### 2.4.8. Calculation of the Thickness of the Circular Rings (e5 Inner) and (e6 Outer)

The vessel is subjected to internal pressure, for which the following formula is applied according to the ASME Code for calculating the wall thickness of the circular rings for both ends of the circular cylinders. See Equation (8)

$$t = \frac{P \times R}{SE - 0.6P} \tag{8}$$

where replacing and calculating, we obtain:

$$t_5 = \frac{0.758 \times 304.8}{93.8 \times 085 - 0.6 \times 0.758} = 2.91 \text{ mm}$$

A thickness of 1/16 inch (1.58 mm) is added as a corrosion factor to select a 3/16-inch plate equivalent to 4.76 mm, as shown in Figure 8.
S = Allowable material stress (93.8 MPa)

E = Welded joint efficiency (0.85)
R = Cylinder radius 304.8 mm y 914.4 mm.

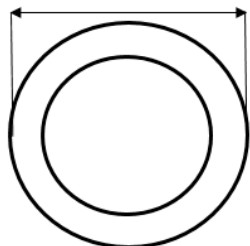
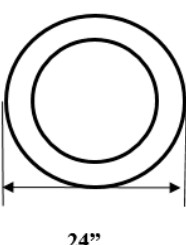

**Figure 8.** Circular covers of vertical cylinders.

$$t_6 = \frac{0.758 \times 914.4}{93.8 \times 0.85 - 0.6 \times 0.758} = \frac{808.63}{79.28} = 10.20 \text{ mm}$$

Considering a corrosion factor of 1/16 inch, a plate is selected of 1/2 inch equivalent to 12.7 mm.

2.4.9. Thickness Calculation (e7)

Figure 9 shows the section where the water reaches its stable value to generate the water vapor and is discharged to the place of use, and Figure 10 shows the Dome side.

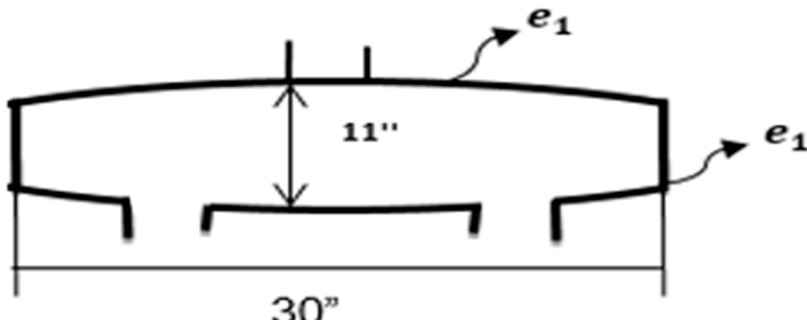

**Figure 9.** Dome formed by elliptical covers to store hot water and steam.

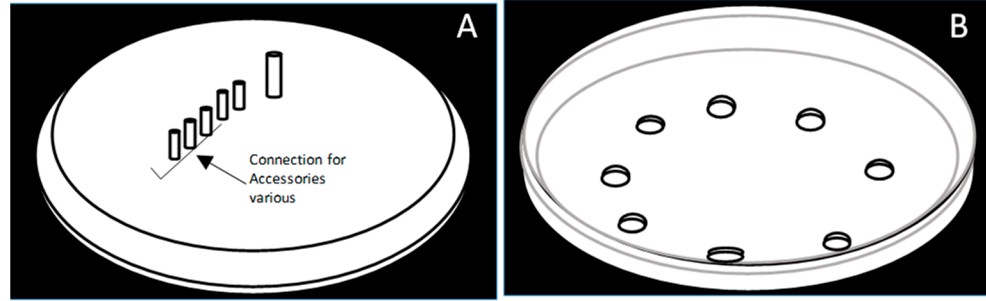

**Figure 10.** The upper side of the Dome (**A**), the Lower side of the Dome (**B**).

The following Equation (9) is used to calculate the thickness of the elliptical lids subjected to internal pressure Pi:

$$T = \frac{P \times D}{2SE - 0.2P} = \frac{0.758 \times 762}{2 \times 93 \times 8 \times 0.85 - 0.2 \times 0.758} = \frac{577.6}{159.3} = 3.63 \text{ mm} \qquad (9)$$

Considering a corrosion factor of 1/16 inch, a plate is selected from 1/4 inch or 6.35 mm
S = Allowable material stress (93.8 MPa)

E = Welded joint efficiency (0.85)
D = Vessel diameter (762.0 mm)
P = Upper vessel internal pressure (0.758 MPa)

Figure 11 shows the parts of the pyroacuotubular (mixed) boiler with each of its components, as described in the figures above.

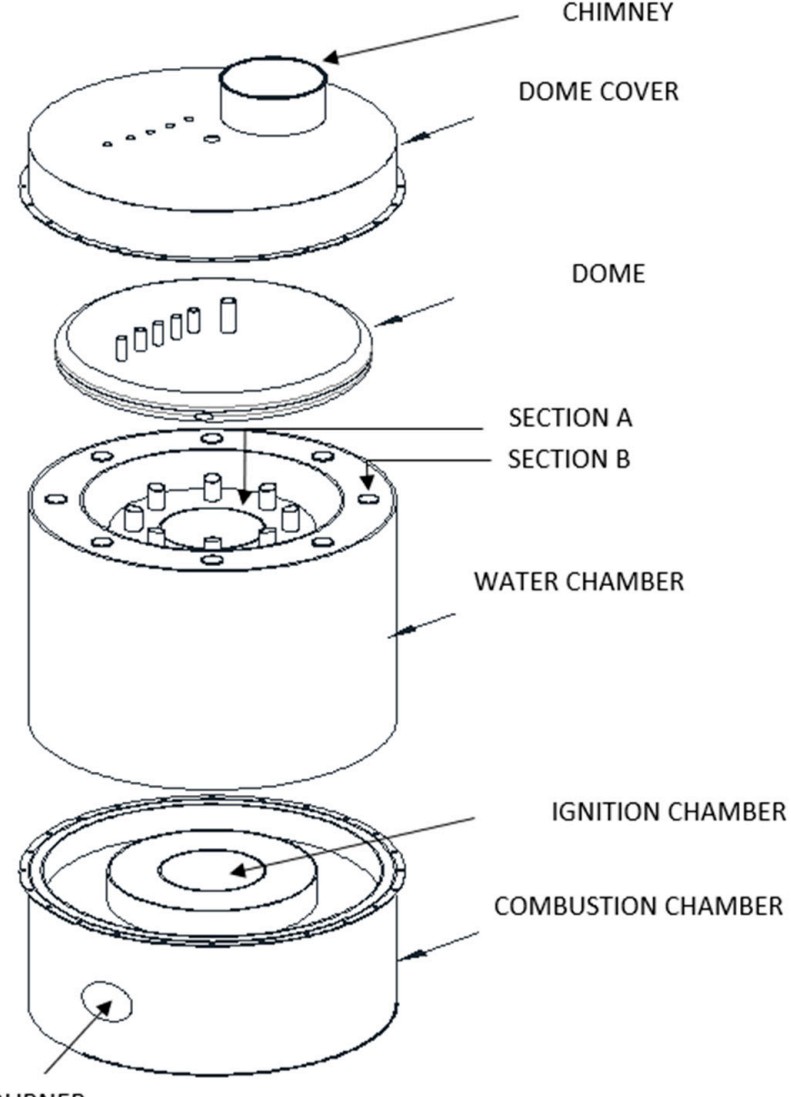

**Figure 11.** Components of the pyroacuotubular (mixed) boiler.

## 3. Results

### 3.1. Calculation of Heat Flows

To achieve this necessary purpose, we will calculate the heat as a function of the time elapsed during the water heating and steam generation process, which will allow us to perform the heat calculations as a flow. Additionally, we will consider the combustion and thermal efficiency factors calculated in [1], as follows: $\eta_{cc} = 99\%$, $\eta_t = 92.4\%$.

The sequence of water heating and steam generation process is showed as follow, Figure 12 shows the water inlet process, Figure 13 shows the water flow in the ignition chamber, Figure 14 shows the action of the combustion gases and Figure 15 the water steam generation.

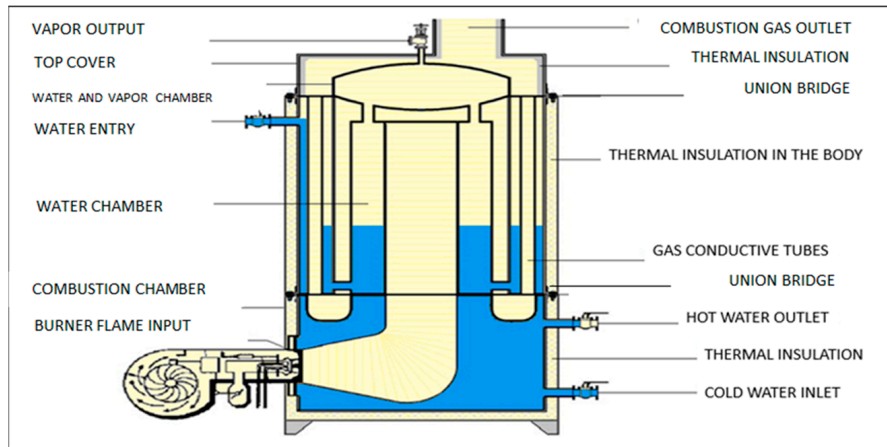

**Figure 12.** Water inlet process.

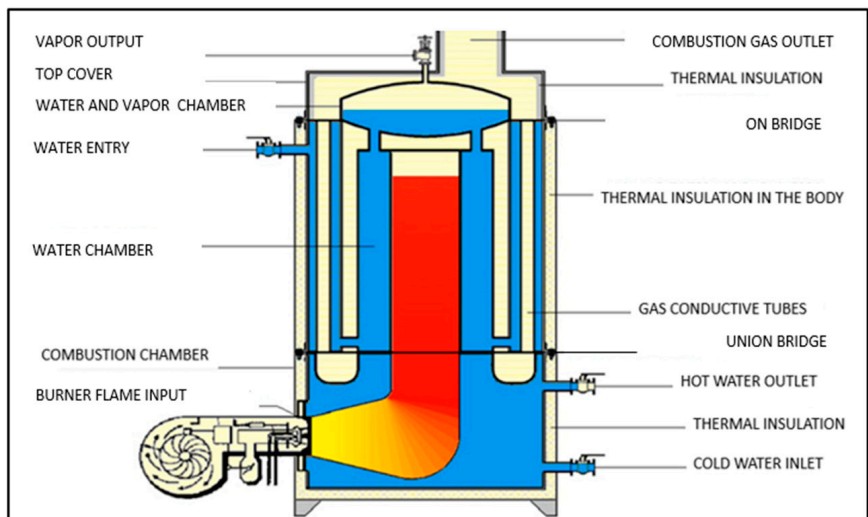

**Figure 13.** The ignition chamber surrounded by water and the water chamber is at its maximum level (blue color), and the burner is on, and the combustion gases are making their first run.

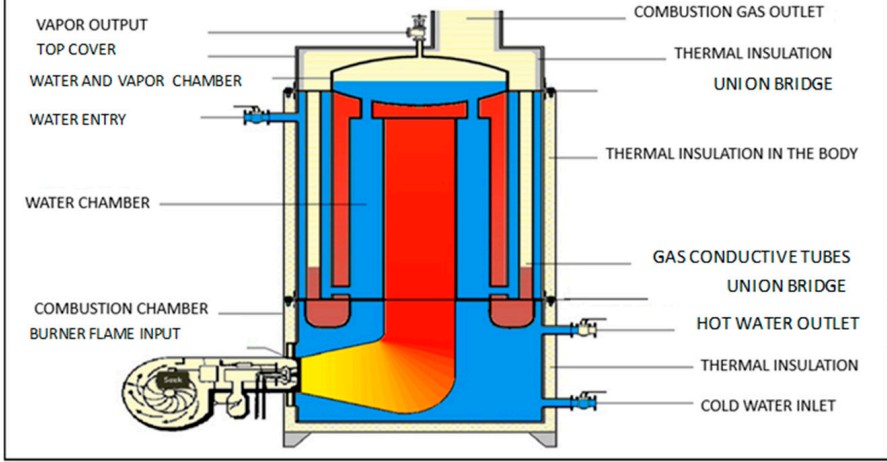

**Figure 14.** The combustion gases have completed their first run and are starting their third.

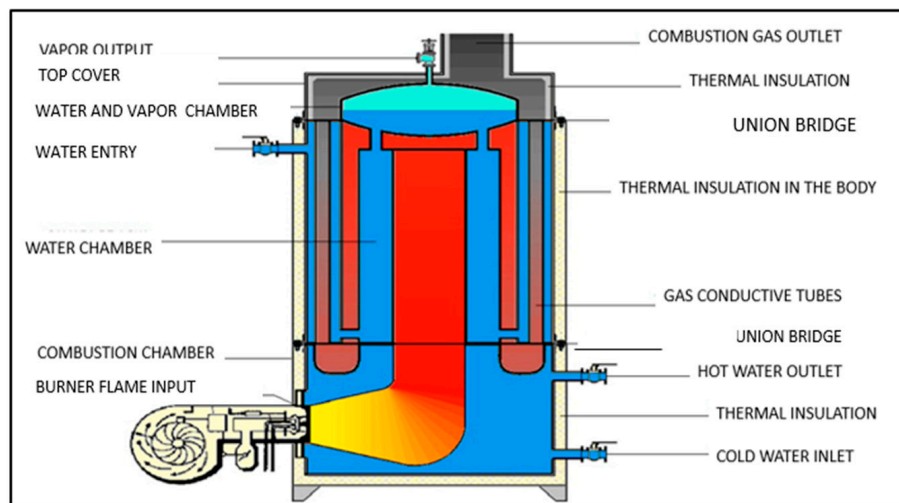

**Figure 15.** The dome shows the water steam (light blue color) and the exit of the hot water in the combustion chamber; in gray and black color, the combustion gases coming out of the chimney are shown.

### 3.1.1. Heat Flow in the Combustion Chamber

According to [1], it has a combustion chamber built internally with insulating material that gives rise to the firebox, all this within an externally insulated metal cylinder, which is considered adiabatic (does not generate heat losses to the external environment); this internal insulating material in this work is replaced by water to be heated, which will involve greater use of the heat generated by combustion, as shown in Figure 4, then we have; as a result, the total mass containing water inside a cylindrical metal mass that is calculated in Equation (10).

$$M_{total} = m\ water + m\ metal \tag{10}$$

$$M_{total} = 489.77 + 193.99 = 683.76\ kg$$

It states that, for heating water, for a total mass of:

$$M_{total} = 1630\ kg\ with\ LPG\ fuel\ (CVF = 52,123\ kJ/kg)$$

Time required, t = 0.55 h (33 min); calculating a new fuel flow if only water is heated in the combustion chamber proportionally for 683.76 kg, 0.23 h is required, therefore, the required heat flow in the combustion chamber is: 164,956.72 kJ.

$$\dot{Q}_{theoretical} = \frac{164,956.72\ kJ}{0.23\ h} = 717,203.13\ kJ/h$$

Then:

$$\begin{aligned}\dot{Q} &= \dot{m} \times \frac{CVF}{\eta_{cc}}\\ &= \frac{717,203.13}{0.99} = 724,447.60\end{aligned} \tag{11}$$

$$\begin{aligned}\dot{Q} &= \dot{m}_c \times \frac{CVF}{\eta_t}\\ &= 724,447.60/0.924 = 784,034.20\end{aligned} \tag{12}$$

$$Q_{total\ real} = 784,034.20\ kJ/h$$

From this value, the fuel mass is calculated as:

$$\dot{m}_c = \frac{Q_{TR}}{CVF} = \frac{784,034.20}{52,123} = 15.04\ kg/h$$

3.1.2. Heat Flow in the Hot Water Boiler

The boiler has a combustion chamber with the firebox surrounded externally by water (see Figure 4), as mentioned above; therefore, all the heat generated in the combustion is used in the water chamber and the boiler dome; we have

Metal mass = 660 kg

*Water mass* = 707.38 kg (Includes water with full-dome)

$$Total\ mass = 1367.38\ kg$$

the heat required as follows:

$$Q = 252,579.12 + 50,685.43$$
$$Q = 303,264.55\ kJ$$

The measured time for this heat is 0.55 h (33 min)

The heat flux is:

$$\dot{Q} = 303,264.55\frac{kJ}{0.55\ h}$$
$$\dot{Q} = 551,390.09\ kJ/h$$

Then:

$$
\begin{aligned}
\dot{Q} &= \frac{\dot{m}_c CVF}{\eta_{cc}} \\
&= \frac{551,390.09}{0.99} = 556,959.68\frac{kJ}{h}
\end{aligned}
\tag{13}
$$

$$
\begin{aligned}
\dot{Q}_{total\ real} &= \dot{m} \times \frac{CVF}{\eta_t} \\
&= 556,959.68/0.924
\end{aligned}
\tag{14}
$$

$$\dot{Q}_{total\ real} = 602,770.21\ kJ/h$$

With this value the fuel flow is calculated as:

$$
\begin{aligned}
\dot{m}_c &= \frac{\dot{Q}_{total\ real}}{CVF} \\[6pt]
&= \frac{602,770.21}{52,123} = 11.56\ kg/h
\end{aligned}
\tag{15}
$$

$$\dot{m}_c = 11.56 kg/h$$

3.1.3. Heat Flow in the Steam Boiler

With the same criteria as in the previous section, it is considered that steam is generated inside the dome, but with a smaller volume of water; the difference in volume inside the dome (total volume of the dome—volume occupied by the water) is called the steam chamber as a vertical boiler with one flue gas pass, which is a percentage of the total volume of the water chamber in the pyroacuotubular (mixed) boiler.

The criterion to be used in this case is similar to that of a water-tube boiler whose steam chamber ranges between 12.5% and 14% of the volume of water stored; the volume of water contained in the combustion chamber is not considered in this Thesis, for the calculation of the steam chamber in the dome for the calculation of heat and combustion flow in the following is used:

Metallic mass = 660 kg

Mass of water = total mass of water—free mass of the steam chamber

$$= 707.38 - 14\% \times (707.38)\ kg$$

$$= 707.38 - 99.03$$

$$Water\ mass = 608.35\ kg$$

The total generation time measured is 0.69 h (41.8 min), which considers the heating time up to 100 °C and the vapor phase change; the heat is obtained.

$$Q_{total} = 252,579.12 + Q_i + Q_A + Q_{domo} \tag{16}$$

$$\dot{Q}_A = \text{Amount of heat utilized for phase change (530 766.43 kJ/h)}$$
$$Q_i = heat\ needed\ to\ heat\ the\ new\ volume\ of\ water\ in\ the\ dome$$
$$volume\ of\ water\ in\ the\ dome\ whose\ volume\ is$$
$$V_i = 39,12\ gal\ (volume\ of\ water\ in\ the\ dome) - 26.19\ gal\ (volume\ of\ 99.03\ kg)$$
$$V_i = 12.92\ gal$$

As a result: $m_i$ = new volume of water in the dome to generate steam
$m_i$ = 48.83 kg

$$Q_i = mi_i * Cp\ (T_f - T_i) \tag{17}$$

$$Q_i = 48.83\ kg \times 4.18 \frac{kJ}{kg} °k \times (373 - 295)\ °k$$

$$Q_i = 15,920.53\ kJ$$

$$Q = Q_{total\ water\ chamber} + Q_{initial\ new\ dome\ volume} + Q_{metal\ dome\ mass}$$

Calculation of the value $Q_i$ results

$$Q_{sub\ total} = 252,579.12 + 15,920.53 + 2770.13$$

$$Q_{sub\ total} = 271,269.78\ kJ$$

This value of the heat considered as heat flow, to heat the water up to 100 °C, as set forth above, has a time of 0.55 h (33 min)

$$\dot{Q}_{sub\ total} = 271,269.78 \frac{kJ}{0.55} h\ = 493,217.78\ kJ/h$$

$$\dot{Q}_{sub\ total} = 493,217.78\ \frac{kJ}{h}$$

For the total heat required for steam generation:

$$\dot{Q}_{total} = \dot{Q}_{sub\ total} + \dot{Q}_A \tag{18}$$

Replacing:

$$\dot{Q}_{total} = 493,217.78 \frac{kJ}{h} + 530,766.43\ \frac{kJ}{h}$$

$$\dot{Q}_{total\ theoretical} = 1,023,984.21\ kJ/h$$

To calculate the theoretical fuel flow of fuel in the Equation:

$$\dot{Q}_T\ =\ \dot{m}_c * P_C$$

$$\dot{m}_c\ =\ Q_T / P_C \tag{19}$$

$$\dot{m}_c = \frac{1,023,984.21}{52,123} = 19.64\ kg/h$$

The $\dot{m}_C$ is calculated considering efficiencies of [1]

$$\eta_{combustion} = 0.99\ y\ \eta_{thermal} = 0.924$$

$$\frac{\dot{Q}_{total\ theoretical}}{\eta_c * \eta_\tau}\ = \dot{Q}_{total\ real} \tag{20}$$
$$= 1,023,984 \times \tfrac{21}{0.99} = 1,034,327.48$$

Final calculation: $\dot{Q}_{\text{total real}} = 1.034,327.48/0.924$, as heat flow results:

$$\dot{Q}_{\text{total real}} = 1,119,402.04 \text{ kJ/h}$$

Actual fuel flow

$$\dot{m}_c = \frac{\dot{Q}_{\text{total real}}}{\text{CVF}} = 1,119,402.04/52,123$$

$$\dot{m}_{comb\ real} = 21.47 \text{ kg/h}$$

3.1.4. Heat in the Boiler for Simultaneous Generation of Steam and Hot Water

The evaluation of this form of utilization of combustion heat and fuel flow is considered in the following ways:

- The efficiencies (combustion and thermal efficiency) are calculated and used;
- The heat source is a single source, uses a single fuel type as in the previous analyses, and is combusted in a burner;
- The heat distribution is simultaneous to heating water and generating steam;
- The steam generation and water heating time will be longer (0.69 h = 41.8 min);
- It will be necessary to recalculate the fuel flow considering the heat requirements established in Table 2.

Table 4 is a summary showing the required heat flows, as well as the total heat required to obtain hot water in the proposed boiler:

**Table 4.** Theoretical heat fluxes to obtain hot water.

| Boiler Components | Heat Fluxes |
|---|---|
| Combustion chamber: total heat | 164,956.72 kJ/h |
| Water chamber: total heat | 252,579.12 kJ/h |
| Dome, metal mass: total heat | 2770.13 kJ/h |
| Dome, new volume of water: total heat | 15,920.53 kJ/h |
| Total heat for heating water | 436,229.50 kJ/h |

From the theoretical heat obtained from Table 2, by applying the heat efficiencies, we will calculate the actual values of the fuel required.

$$\dot{Q}_{\text{total theoretical water}} = 436,226.50 \text{ kJ/h}$$

Theoretical heat flow $\dot{Q}_{\text{theoretical water}} = \frac{436,226.50 \text{ kJ/h}}{0.69}$

$$\dot{Q}_{\text{theoretical water}} = 632,212.32 \text{ kJ/h}$$

Fuel flow:

$$\dot{m}_{\text{Fuel flow}} = 632,212.32 \text{ kJ/h}/52,123 \text{ kJ/kg} = 12.13 \text{ kg/h}$$

For water heating: The fuel flow is as follows

$$\dot{m}_c = 12.13 \text{ kg/h}$$

The actual heat as a flux considers the calculated efficiencies by [1]

$$\eta_{combustion} = 0.99 \quad \eta_{termica} = 0.924$$

Applying the water heating efficiencies in the calculation, the following result is obtained

$$\dot{Q}_{\text{real Sub total agua}} = 632,212.32 \text{ kJ/h}/0.99 = 638,598.30 \text{ kJ/h}$$

$\dot{Q}_{\text{real total agua}} = 638,598.30\frac{\frac{\text{kJ}}{\text{h}}}{0.924} = 691,123.70\frac{\text{kJ}}{\text{h}}$, then the Total actual heat flux required to heat water is obtained

$$\dot{Q}_{\text{real total agua}} = 691,123.70 \text{ kJ/h}$$

Necessary fuel flow results from:

$$\dot{m}_c = \frac{691,123.70 \text{ kJ/h}}{52,123 \text{ kJ/kg}}$$

$$\dot{m}_{c\ real} = 13.26 \text{ kg/h}$$

The fuel flow necessary to heat water up to 100 °C in the whole boiler.

The heat required ($\dot{Q}_A$) to generate steam in the corresponding section of the boiler (dome) is the value of $\dot{Q}_A$ calculated above for 150 kW.

$$\dot{Q}_A = 530,766.43 \text{ kJ/h}$$

Finally, the total theoretical heat required will be:

$$\dot{Q}_{\text{total teorico}} = \dot{Q}_{\text{teorico agua}} + \dot{Q}_A \tag{21}$$

Replacing

$$\dot{Q}_{\text{total required}} = 632,212.32 + 530,766.43$$

$$\dot{Q}_{\text{total theoretical necesary}} = 1,162,978.75 \text{ kJ/h}$$

Calculating the actual values of heat and fuel, applying combustion and thermal efficiencies.

$$\dot{Q}_{\text{Sub total actual steam and water}} = \frac{1,162,978.75 \text{ kJ/h}}{0.99}$$
$$= 1,174,726.01 \text{ kJ/h}$$

$$Q_{\text{total actual steam and water}} = 1,174,726.01\frac{\text{kJ}}{\text{h}}/0.924$$
$$= 1,271,348.49 \text{ kJ/h}$$

The actual fuel flow is: $\dot{Q} = \dot{m}_c * \text{CVF}$

$$\dot{m}_c = \dot{Q}/\text{CVF}$$
$$= 1,271,348.49 \text{ kJ/h}/52,123 \text{ kJ/kg} \tag{22}$$

$$\dot{m}_{\text{comb. real steam and water}} = 24.39 \text{ kg/h}$$

Tables 5 and 6 show the summary of all the previous calculations related to the simultaneous generation of hot water and steam with its different alternatives, as well as the fuel consumption for a given time.

**Table 5.** Results of fuel consumption and heat requirements.

| Type of Fluid to Be Generated | Generation Time: h (min) | Theoretical Heat Required kJ/h | Actual Heat Required kJ/h | Fuel Flow ($\dot{m}_c$) kg/h |
|---|---|---|---|---|
| Hot water in the combustion chamber | 0.23(13.8 min) | 717,203.13 | 784,034.20 | 15.04 |
| Only hot water in the boiler | 0.55 (33 min) | 551,390.09 | 602,770.21 | 11.56 |
| Only steam in the boiler | 0.69 (41.8 min) | 1,023,984.21 | 1,119,402.04 | 21.47 |
| Simultaneous steam and hot water in the boiler | 0.69 (41.8 min) | 1,162,978.75 | 1,271,348.49 | 24.39 |

**Table 6.** LPG fuel consumption.

| Type of Fluid to Be Generated | Consumption kg/Time | | | |
|---|---|---|---|---|
| | kg/h | 24 h/Day | 730 h/Month | 8760 h/Year |
| Hot water only | 11.56 | 277.44 | 8438.8 | 101,265.6 |
| Steam only | 21.47 | 515.23 | 15,673.1 | 188,077.2 |
| Parallel generation of hot water and steam | 33.03 | 792.72 | 24,111.9 | 289,342.8 |
| Simultaneous steam and hot water generation | 24.39 | 585.36 | 17,804.7 | 213,656.4 |

Table 6 shows the individual values of fuel consumption and heat per hour; for this thesis, we will analyze the different alternatives, i.e., only hot water, steam, and as fuel consuming equipment operating in parallel, and finally, the simultaneous generation of hot water and steam, these values would be analyzed, considering the operation of 24 h/day for a total of 8 760 h/year, as shown in Table 5.

Table 6 shows that the fuel consumption for the simultaneous generation of steam and hot water is lower than the fuel consumption of the equipment operating in parallel, i.e., one generating hot water and the other steam. However, another important factor to consider within the objectives of the thesis is the production of combustion gases that affect the environment, such as carbon dioxide ($CO_2$) and carbon monoxide (CO), to determine the values of these gases to the environment, it is necessary to calculate from a chemical reaction the quantities emitted according to that established by [1] in his master's thesis for the fuel used (LPG).

Therefore, only one burner was used for the tests, resulting in the following flue gas values:

$$CO_2 = 12.4\%, \ CO = 0.06\%, \ O_2 = 2.4\% \ N_2 = 85.14\%$$

$$T_O = 22\,°C \ T_{chimney \ gases} = 183\,°C$$

The chemical formula of the fuel: $C_3H_8$
The chemical formula of the fuel (M) = 44
To calculate $\dot{m}_{co_2}$, $\dot{m}_{co}$, it is necessary to calculate
Theoretical air–fuel ratio ra/ct
Actual air-fuel ratio ra/cr
Excess air in percent %e
These results will be obtained from the chemical reaction of the fuel with the following procedure:
The chemical reaction Equation expressed per mole of fuel

$$C_3H_8 + (O_2 + 3.76\,N_2) \rightarrow 12.4CO_2 + 0.06CO + 2.4O_2 + 85.14N_2 + H_2O \qquad (23)$$

Resolved

$$AC_3H_8 + B(O_2 + 3.76N_2) \rightarrow 12.4CO_2 + 0.06CO + 2.4O_2 + 85.14N_2 + EH_2O \qquad (24)$$

Balancing by elements

$$C : 3A = 12.4 + 0.06$$
$$A = 4.15$$

$$H : 8A = 2E, \text{ replacing } 8 \times 4.15 = 2E$$
$$E = 16.61$$

$$O : 2B = 12 \times 4 \times 2 + 0.06 + 2.4 \times 2 + E$$
$$= 29.66 + E \text{ replacing}$$
$$2B = 29.66 + 16.61$$
$$2B = 46.27$$
$$B = 23.135$$

The Equation (24) is as follows

$$4.15C_3H_8 + 23.135(O_2 + 3.76N_2) \rightarrow 12.4CO_2 + 0.06CO + 2.4O_2 + 85.14N_2 + 16.61H_2O \tag{24}$$

To calculate the actual air-fuel ratio is as follows.

$$r_{a/c \; real} = \frac{real \; air \; mass}{fuel \; mass}$$
$$= \frac{23.135(16 \times 2 + 3.76 \times 14 \times 2)}{4.15(3 \times 12 + 8)} = \frac{3175.97}{182.6}$$

$$r_{a/c \; real} = 17.39$$

The theoretical chemical reaction to be calculated is given from the following Theoretical Equation (25).

$$C_3H_8 + B(O_2 + 3.76N_2) \rightarrow XCO_2 + 4H_2O + ZN_2 \tag{25}$$

Balanced:

$$C : X = 3$$
$$H : 8 = 24; Y = 4$$
$$O : 2B = 2X + Y; \; B = 5$$
$$N : 2 \times 3.76 \times B = 2Z; \; Z = 18.8$$

The resulting Equation is:

$$C_3H_8 + 5(O_2 + 3.76N_2) \rightarrow 3CO_2 + 4H_2O + 18.8N_2$$

$$r_{a/c \; terico} = \frac{theoretical \; air \; mass}{fuel \; mass}$$

$$= \frac{5(16 \times 2 + 3.76 \times 14 \times 2)}{(3 \times 12 + 8)} = \frac{686.4}{44} = 15.6$$

$$ra/c \; _{theoretical} = 15.6$$

Excess air is calculated as:

$$\%e = \frac{A_r - A_t}{A_t} 100 = \frac{17.39 - 15.6}{15.6} = 11.47$$

$$\%e = 11.47$$

The resulting mass value of $CO_2$ (carbon dioxide) and $CO$ (carbon monoxide) is determined from the actual chemical reaction Equation (26), which is shown as follows.

$$4.15C_3H_8 + 23.135(O_2 + 3.76N_2) = 12.40CO_2 + 0.06CO + 2.4O_2 + 85.14N_2 + 16.61H_2O \tag{26}$$

By performing a mass balance, the result is

$$182.6 + 23.135(32 + 105.28) = 545.6 + 1.68 + 76.8 + 2383.92 + 298.98 \qquad (27)$$

The result : $3358.57 = 3306.98$

The difference in masses is due to the non-combustion gases such as helium and argon and the presence of nitrogen, which is an inert gas at low combustion temperatures, then the mass balance is taken into account:

$$CO_2 = 545.6 \text{ kg} CO = 1.68 \text{ kg}$$

It is considered that these values are emitted to the environment in the alternative cases mentioned above, for which the necessary values are calculated as follows in the function of the time considered in Table 4.

The mass of gases emitted to the environment is obtained; from this Equation, $CO_2$ and CO are used, which are the gases that affect the environment with the greatest intensity, and the following results are obtained by calculating the mass flow as a function of time, depending on whether it is hot water or steam:

To generate hot water only:

$$CO_2 = 545.6/0.55 = 992 \text{ kg/h}$$
$$CO = 1.68/0.55 = 3.054 \text{ kg/h}$$

To generate only water vapor:

$$CO_2 = 545.6/0.69 = 790.739 \text{ kg/h}$$
$$CO = 1.68/0.69 = 2.43 \text{ kg/h}$$

For a parallel generation of hot water and steam

$$CO_2 = 992.0 + 790.72 = 1782.72 \text{ kg/h}$$
$$CO = 3.054 + 2.43 = 5.48 \text{ kg/h}$$

For a simultaneous generation of steam and hot water:

$$CO_2 = 545.6/0.69 = 790.72 \text{ kg/h}$$
$$CO = 1.68/0.69 = 2.43 \text{ kg/h}$$

Setting only the amount of carbon emitted to the environment is obtained from the actual Equation:

$$De : 12.4C_2 = 12.4(C + O_2)$$

Only coal results:

$$C_1 = 12.4 \times 12 = 148.40 \text{ kg}$$
$$0.06 \, CO = 0.06(C + O)$$

Coal results in

$$C_2 = 0.06 \times 12 = 0.72 \text{ kg}$$

Total carbon emitted is:

$$C_1 + C_2 = 148.40 + 0.72 = 149.52$$
$$C = 149.52 \text{ kg}$$

A time factor is applied to the carbon value resulting from the calculation to obtain the mass flow for a second calculation for each type of fluid considering the generation alternatives, as follows:

To generate hot water only $C : 149.52 / 0.55 = 271.85 \text{ kg/h}$

To generate water vapor only:

$$C : 149.52/0.69 = 216.69 \text{ kg/h}$$

For a parallel generation of hot water and steam

$$C : 271.85 + 216.69 = 488.54 \text{ kg/h}$$

For simultaneous steam and hot water generation

$$C : 149.52/0.69 = 216.69 \text{ kg/h}$$

Tables 7 and 8 show the hourly, monthly, and annual values of flue gas emissions and carbon emissions to the environment.

**Table 7.** Amount of combustion gases emitted to the environment.

| Fluid Type | Emission | Combustion Gases Emitted into the Environment | | | | |
| | | kg/h | kg/Day | kg/Month | kg/Year | % Reduction |
|---|---|---|---|---|---|---|
| Hot water | $CO_2$ | 992.00 | 23,808.00 | 714,240.00 | 8,689,920.00 | 44.35 |
| | CO | 3.05 | 73.30 | 2198.88 | 26,753.04 | 44.27 |
| Steam | $CO_2$ | 790.72 | 18,977.28 | 569,318.40 | 6,926,707.20 | 55.65 |
| | CO | 2.43 | 58.32 | 1749.60 | 21,286.80 | 55.66 |
| Parallel generation | $CO_2$ | 1782.72 | 42,785.28 | 1,283,558.40 | 15,616,627.20 | 0.00 |
| | CO | 5.48 | 131.52 | 3945.60 | 48,004.80 | 0.00 |
| Simultaneous generation | $CO_2$ | 790.72 | 18,977.28 | 569,318.40 | 6,926,707.20 | 55,65 |
| | CO | 2.43 | 58.32 | 1749.60 | 21,286.80 | 55.66 |

**Table 8.** Carbon emissions to the environment by fluid type.

| Mass/Time | Hot Water | Steam | Parallel Generation | Simultaneous Generation |
|---|---|---|---|---|
| 1 h (kg/h) | 271.85 | 216.69 | 488.54 | 216.69 |
| 24 h (kg/day) | 6254.4 | 5200.56 | 11,724.96 | 5200.56 |
| 730 h (kg/month) | 199,450.50 | 178,623.7 | 356,634.2 | 158,183.7 |
| 8760 h (kg/year) | 2,381,406 | 1,898,204.4 | 4,279,610.4 | 1,898,204.4 |

Table 8 shows the amount in kg of gases emitted to the environment per hours of operation and the percentage of reduction.

## 4. Discussion

In the new design of the boiler for the simultaneous generation of hot water and steam in the present research work, it was necessary to consider the development of a second generation or technological improvement of a first research work called "Dimensioning and construction of a three-pass multipurpose mixed vertical boiler (pyroacuotubular mixed) to optimize the level of thermal efficiency with alternative fuels", the design of the pyroacuotubular (mixed) boiler is considered an innovative design, which takes advantage of the boiler's combustion chamber to generate hot water and steam simultaneously; this type of generation occurs in single equipment to reduce the emission of greenhouse gases, significantly contributing to the decarbonization of the environment. This innovative design solves a global technological problem because after searching for similar options, it was not possible to find a boiler on the market that is pyroacuotubular (mixed), vertical with three gas passages, and that simultaneously generates hot water and steam.

This innovative proposal for reducing greenhouse gas emissions and reducing the emission of carbon into the environment correlates with the modification in the design of a boiler made in the design by Wejkowsky [16] in 2016 in his research work called "Triple fine tubes increasing efficiency decreasing $CO_2$ pollution of a steam boiler" in which he

manages to increase thermal efficiency by 90%, as well as reducing the temperature of combustion gases from 158.8 °C to 138 °C and also reduce the emission of $CO_2$ up to 826 tons per year, similarly [17] in 2014, a scientific article in which using a neural network system and based on a study of the environment predicts the behavior of a coal-fired power plant that has three water-tube boilers, and by this method, proposed to reduce the emission of $CO_2$ by 5, 628 tons per year.

In this research, the combustion chamber is observed where the water is heated externally to the ignition chamber or hearth; likewise, the dome containing the hot water for steam generation is observed. Analyzing fuel consumption and heat required for the generation of hot water and steam separately and simultaneously, it is observed that the sum of fuel consumption for parallel generation of hot water and steam is much higher than the consumption of simultaneous generation. Likewise, the heat required for parallel generation is much higher than that required for simultaneous heating of hot water and steam in the theoretical and real cases, as is confirmed in [5,15,32].

As seen in the tables, the reduction in greenhouse gases emitted to the environment decreases significantly in the annualized calculation for both $CO_2$ and CO emissions, thus demonstrating that the design of the new type of boiler is favorable for the objectives of Sustainable Development.

Economically, this boiler has a very interesting potential for market presence, considering this prototype a second invention; the first one is commented on in the work of [1]. The differential of this second invention is the production of hot water and steam. Environmentally, emissions and waste are reduced, and their temperature is reduced, generating better use of combustion gases and a lower environmental impact than other industrial boilers. Socially, the contribution of this type of design generates a product responsibility, by guaranteeing better working conditions for the operators who work with this type of boiler, and an environment [12] with greater comfort, by reducing the health risks that these boilers can generate.

According to the guidelines outlined by ECLAC within the policies to close the gaps of Sustainable Development [4–6], it is required to implement Decarbonization, Structural Change, and Technical Progress, and precisely this research work with the contribution of the design of the boiler for simultaneous generation of hot water and steam seeks to collaborate with decarbonization and contribute to technical progress, which is the basis for structural change, as can be seen in Figure 16.

**Figure 16.** Economic Commission for Latin America and the Caribbean (ECLAC).

## 5. Conclusions

Based on the research of [1], in which his new type of vertical mixed boiler with three passes or gas paths through its heat transfer surface for multipurpose use with alternative

fuels optimizes thermal efficiency; by producing steam and hot water, contemplating the possibility of using alternative fuels such as diesel oil, natural gas and Liquefied Petroleum Gas (LPG), this new design reduces the emission of combustion gases since by introducing a volume of water in the combustion chamber surrounding the ignition chamber, energy is used, due to the fact that the combustion chamber contains an ignition chamber inside and its external surface will be surroundesd by water that recirculates in order to take advantage of the combustion heat, obtaining hot water. The new proposal compared carbon emissions to the environment by fluid type; hot water, water steam, and parallel generation show that simultaneous generation generates greater benefits in relation to environmental pollution and the findings of this research lead us to conclude that the new design of the pyroacuotubular (mixed) boiler makes it possible to simultaneously generate hot water and water steam, and also to reduce the temperature of the gases emitted by increasing the thermal efficiency of the boiler, compared to other industrial boilers, with better use of the combustion gases, reducing the emission temperature of these gases, which, in turn, significantly reduces the emission of greenhouse gas pollutants, contributing to reducing global warming and environmental pollution. On the other hand, previous studies of the modification made by [16] where the modified water-tube boilers are surpassed in efficiency by the present work, as well as [17], the efficiency achieved in the water-tube boilers shows that it is possible to modify it to achieve a new design, which is ratified and improved with the proposed design. Future prospects are encouraging, considering that it is possible to improve the design by applying other methods and techniques based on thermodynamic laws.

Sustainable Development requires technological innovations, as is the case of the design of the water-tube pyroacuotubular (mixed) boiler for a simultaneous generation, because it reduces the emission of greenhouse gases contributing to decarbonization and falls within the objectives of the 2030 Agenda, such as goals 9, 7, and 11 by having a single heat source such as combustion to obtain two types of products that can be used as an energy source of the second category in a block diagram, therefore, it has an important influence on Sustainable Development.

**Author Contributions:** This research was directed and coordinated by D.A.V. as group director investigator; D.A.V., D.E. and C.R. provided methodological and technical conceptual guidance for all aspects of the research. The methodology was planned and executed, and the data analyzed with D.A.V. carried out and analyzed the optimal performance of the design of boiler experiments and construction contributed to the technological development of the design. C.R. writing—reviewing and editing, D.E. contributed with visualization and supervision of the project. D.A.V., D.E. and C.R. contributed to funding the research. The manuscript was written by D.E. and commented on by all authors. All authors have read and agreed to the published version of the manuscript.

**Funding:** This research received no external funding. Own author's resources financed the research.

**Acknowledgments:** We would like to express our special thanks and gratitude to Duilio Aguilar and to the colleagues who gave us the dsgolden opportunity to do this wonderful project on the topic of the vertical boiler with three gas passes, which also helped us in doing a lot of group research on universities as UNI, UNFV, and UNMSM.

**Conflicts of Interest:** The authors declare no conflict of interest.

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
