# Peer review of "Design of a Pyroacuotubular (Mixed) Boiler for the Reduction of Flue Gas Emissions through the Simultaneous Generation of Hot Water and Water Steam"

_fluids, doi:10.3390/fluids7090312_

Round 1

Reviewer 1 Report

Review (recommended major revision)

Authors presented a novel pyro-tube boiler design concept that reduces the emission of produced combustion gases by producing regulated hot H2O or steam simultaneously. All necessary methods for the design, modification and analytical predictive calculation of heat are presented.

***attached review file***

Author Response

Response to Reviewer 1 Comments

Thank you for your reviewers' comments concerning our manuscript entitled “Pyro and Aquotubular Boiler for Reducing Combustion Gas Emissions Through Simultaneous Generation of Hot Water And Steam For Sustainable Development”. Those comments are all valuable and very helpful for revising and improving our manuscript, as well as the important guiding significance to our research. We have studied the comments carefully and have made corrections which we hope to meet with approval. Revised portions are added to the manuscript. The main corrections in the paper and the response to your comments are as flowing:

  • Provide a more appropriate title as it is not clear if the work is a simulation/concept or an already built prototype. The title was given in very general terms. It says nothing about what was done in the paper (simulation), what combustion gases are considered, and what type of technology/industry it is intended for.

Thank you for this valuable suggestion. We have reexamined the title in accordance with your comments a) and b), and the authors have decided to modify the title as follows

  • “Design of a fire tube-water tube (mixed) boiler for the reduction of combustion gas emissions through the simultaneous generation of hot water and steam”
  • Leave out trivial information.
  • Information in all paper was revised and updated leaving out the trivial information
  • Briefly explain how this work differs from previously published work.
  • The difference is that in the previous work, hot water and steam are generated alternately, while in the current design, hot water, steam, and thermal oil are generated simultaneously.
  • State specific objectives and highlight them.

In order to answer the review comment the research question was added and specific objectives have been set

  • How to improve the thermal efficiency of a boiler by reducing the emission of combustion gases in the generation of hot water and steam? and to achieve the goals of the research, specific objectives were defined that will drive the research to narrow the focus and guide the research process, in that sense the following specific objectives have been set:

- Simultaneously generate water and steam

- To reduce the temperature of the gases emitted

- To reduce the emission of greenhouse gas pollutants.

  • Have the English edited very carefully (subscript/superscript).
  • The English was edited very carefully in subscript/superscript the errors as Ft2 (Ft2), CO2 (CO2 ), and others were corrected.

  • Highlight key findings in the conclusions and compare them to other work. In general, expand the discussion in this regard.

Key findings consistent with the proposed specific objectives were included as shown.

Contrasting the most important results of the present research is considered the previous work of Aguilar[1] in which his new type of vertical mixed boiler with three passes or gas paths through its heat transfer surface for multipurpose use with alternative fuels optimizes thermal efficiency; by producing steam and hot water, contemplating the possibility of using alternative fuels such as diesel oil, natural gas and Liquefied Petroleum Gas (LPG), the proposal of the referred research is improved with a design that simultaneously generates hot water and steam, reducing the emission of combustion gases and delivering thermal oil in an alternative way,

This new design reduces the emission of combustion gases since by introducing a volume of water in the combustion chamber surrounding the ignition chamber, energy is used, due to the fact that the combustion chamber contains an ignition chamber inside and its external surface will be surrounded by water that recirculates in order to take advantage of the combustion heat, obtaining hot water. Likewise, the combustion gases are retained due to the shape of the combustion chamber, taking advantage of the combustion heat transferred by the gases, which generates a higher heating rate of the water and lower fuel consumption. In the case of steam generation, the dome located in the upper part of the boiler takes advantage of the heat generated by the combustion gases, obtaining a higher steam generation speed, so that the emission temperature of the combustion gases is reduced and the thermal efficiency of the fire tube-water tube (mixed) boiler is also increased.

The sum of the heat required to obtain only hot water and only steam in parallel for the same power of the unit is a higher value than that of the simultaneous generation of hot water and steam, which has consequences that the amount of fuel consumed to generate hot water and steam also has a lower value as shown by the calculations obtained. The modification made by Wejkowski [16] where he modifies a water-tube boiler is surpassed in efficiency by the present work as well as Strusnik [17] the efficiency achieved in water-tube boilers shows that it is possible to modify to achieve a new design which is ratified and improved with the proposed design. For future projects, we are considering working on a design with thermal oil to increase the temperature and reduce the pressure, very applicable in the textile industry.

  • Remove less important and unrelated or less related references, and ensure that all references are cited and arranged in the order required by the journal.

  • The less important and unrelated or less related references were eliminated and all references were cited and arranged in the order required by the journal.
  • Avoid mass citations whenever possible, for example [2-5].
  • The mass citations were avoid
  1. “There is a growing development of technology dedicated to controlling pollutant gas emissions.” – There is no relevant mentioning, how this is mitigated nowadays, as well as, how emission treatment technologies are being developed, whether CO reduction or NOX decomposition.

  • Articles have been found on how to capture CO2, not exactly how to mitigate it, however, references are being added related to oxyfuel, green oxygen, biofuel technologies as well as research is being done on generating biofuel to lessen the effect of fossil fuels with references
    [4][5][6][10].

  1. There are some discrepancies when comparing Table 4 and Table 6. Table 4 calculates 730 hours for a month, while Table 6 gives 720 hours for a month of 30 days. While you could take 30 days and 10 hours to get 730 hours for the LPG consumption calculation, I think this is an error. Please correct the error or explain in detail the time used for the calculations.

  • Thanks for your comment, there was a digitation error that was corrected, the right value is 730 h/month

  1. Figures should generally be referenced if they are borrowed from another work. If the Figures are the original, I think they should be of better quality.

  • Figures were improved to a better quality.

  1. Units should be consistent throughout the manuscript. For example, BHP and psi should be corrected in the manuscript. No interested reader will have to do the calculations himself.

  • The units throughout the manuscript were revised and corrected

  1. As mentioned in the general comments, there are many subscript and superscript errors in the text and equations that make the manuscript difficult to read. Correct the errors to make it easier to read and understand the proposed work.
  • The English was edited very carefully in subscript/superscript the errors of Ft2 (Ft2), CO2 (CO2 ), and others were corrected

  1. The conclusion lacks a detailed comparative discussion with references to similar studies using similar methods. Highlight key findings and advance the discussion by providing future perspectives.

  • The sum of the heat required to obtain only hot water and only steam in parallel for the same power of the unit is a higher value than that of the simultaneous generation of hot water and steam, which has consequences that the amount of fuel consumed to generate hot water and steam also has a lower value as shown by the calculations obtained. The modification made by Wejkowski [16] where he modifies a water-tube boiler is surpassed in efficiency by the present work as well as Strusnik [17] the efficiency achieved in water-tube boilers shows that it is possible to modify to achieve a new design which is ratified and improved with the proposed design. For future projects we are considering working on a design with thermal oil to increase the temperature and reduce the pressure very applicable in the textile industry.

Reviewer 2 Report

In the submitted manuscript, the authors deal with the current topic of reducing CO2 emissions or decarbonizing the hot water and steam production process. I like the processing of the topic, but I still have a few comments.

In the Abstract, check the numerical values, including separators. In the text, a decimal point and a comma are often mixed in the numerical designation throughout the text.

It is necessary to use only the SI system of units in the description.

Chapter 2.2 – the beginning is in another language, it needs to be translated into English

It is necessary to check the correctness of the entry of units, then it is not clear what it is or it seems quite strange, e.g. "MPab", "MP", "Mpa".

Enter the temperature only in "°C" and not in "degrees C" or "°F"!

Similarly, the place of "kJ" is wrongly given in the manuscript in several places of "kj".

Please check the individual calculation records because there are some errors.

In Chapter 4, "CO2" is replaced by "C02" and similarly "CO" is replaced by "C0".

Further, in chapter 4, there is a graph mentioned in the last sentence, but the graph is missing.

Furthermore, I suggest adding Nomenclature to the manuscript, for a better overview of the used variables and designations.

Author Response

Response to Reviewer 2 Comments

 Thank you for your reviewers' comments concerning our manuscript entitled “Pyro and Aquotubular Boiler for Reducing Combustion Gas Emissions Through Simultaneous Generation of Hot Water And Steam For Sustainable Development”. Those comments are all valuable and very helpful for revising and improving our manuscript, as well as the important guiding significance to our research. We have studied the comments carefully and have made corrections which we hope to meet with approval. Revised portions are added to the manuscript. The main corrections in the paper and the response to your comments are as flowing:

  1. In the Abstract, check the numerical values, including separators. In the text, a decimal point and a comma are often mixed in the numerical designation throughout the text.

The summary was corrected to avoid mixing the decimal point and comma in the numerical designation.

Environmental protection is a continuous challenge that requires innovating the combustion process of boilers that emit polluting gases. This research proposes a novel fire tube-water tube (mixed)  boiler design that reduces the emission of combustion gases through the simultaneous generation of hot water and steam. The applied methodology considers the dimensioning-construction, modification and analytical calculation of water volume, metallic masses, heat for hot water and steam generation, and combustion gases. The Ganapaty method of heat transfer is applied prioritizing the velocity of gas displacement, the pressure drop along the pipe and its application on surfaces. In the parallel generation of hot water and steam, a mass of CO2 (1 782,72 kg/h) and CO (5,48 kg/h) was obtained, these masses were compared with the results of the proposed design, obtaining a reduction in the mass of gases emitted to the environment in: hot water CO2 (44,35%) and CO (44,27%); steam CO2 (55,65%) and CO (55,66%). A significant reduction was achieved in the simultaneous generation of hot water and steam, compared to the individual generation, which shows that the simultaneous generation of the fire tube-water tube boiler reduces the emission of combustion gases.

  1. It is necessary to use only the SI system of units in the description.

The observation has been taken into account and the international system of units is being considered.

  1. Chapter 2.2 – the beginning is in another language, it needs to be translated into English

Section 2.2 was corrected as follows

2.2 Procedures

The procedure evaluates the stages of Dimensioning-construction, modification, analytical calculation, and comparative evaluation, as shown below in Figure 3.

  1. It is necessary to check the correctness of the entry of units, then it is not clear what it is or it seems quite strange, e.g. "MPab", "MP", "Mpa".

The correction of the unit entry was corrected throughout the manuscript.

  1. Enter the temperature only in "°C" and not in "degrees C" or "°F"!

The temperature was changed only in "°C" as suggested

  1. Similarly, the place of "kJ" is wrongly given in the manuscript in several places of "kj".

The error of "kj" was corrected with "kJ" thought all the manuscript

  1. Please check the individual calculation records because there are some errors.

The individual calculation was verified and corrected

  1. In Chapter 4, "CO2" is replaced by "C02" and similarly "CO" is replaced by "C0".

The error of "C02" and similarly "C0" was corrected with “CO2” and “CO".

  1. Further, in chapter 4, there is a graph mentioned in the last sentence, but the graph is missing.

The  graph mentioned in the last sentence, was added.

  1. Furthermore, I suggest adding Nomenclature to the manuscript, for a better overview of the used variables and designations.

The nomenclature to the manuscript was added.

Round 2

Reviewer 1 Report

Review (recommended major revision)

Comments were not addressed appropriately. For example, specifically, 1, 2, 4...

Author Response

Response to reviewer comments

 Comments were not addressed appropriately. For example, specifically, 1, 2, 4...

We would like to start by thanking the reviewer for this valuable feedback. We acknowledge the importance of the reviewer's questions and recommendations and would like to address them point-by-point. The questions have been analyzed and discussed in depth and we hope that the answers will be as expected.

  1. Some general comments:
  • Provide a more appropriate title as it is not clear if the work is a simulation/concept or an already built prototype. The title was given in very general terms. It says nothing about what was done in the paper (simulation), what combustion gases are considered, and what type of technology/industry it is intended for.

According to your suggestion, the authors decided to change the title as follows, which we consider appropriate.

“Design of a pyroacuotubular (mixed) boiler for the reduction of flue gas emissions through the simultaneous generation of hot water and water steam”

  • Leave out trivial information.

Trivial information was left out, but the information was also expanded at the suggestion of the other reviewer.

  • Briefly explain how this work differs from previously published work.

The difference is that in the previous work, hot water and steam are generated alternately, while in the current design, hot water, and water steam are generated simultaneously. It is important to clarify that the first study was an invention and this second one is a utility model that improves on the first one in the simultaneous generation of hot water, and water steam.

  • State specific objectives and highlight them.

Specific objectives have been set:

- Generate simultaneously hot water and water steam, this objective is achieved with a new design of the 3-pass pyroacuotubular (mixed) boiler with LPG fuel that establishes the heat needs required by the metallic mass and water to generate hot water and steam simultaneously based on the Zero Law of Thermodynamics to reach thermal equilibrium.

- Reduce the temperature of the gases emitted, by increasing the thermal efficiency of the pyroacuotubular (mixed) boiler, a better use of the combustion gases is achieved by generating hot water and steam at a higher speed reducing the flue gas emission temperature of the flue gas.

- Reduce the emission of greenhouse gas pollutants, the reduction of the gas temperature generates a better use of the combustion gases and at the same time a lower environmental impact, compared to other industrial boilers, contributing to reducing global warming and environmental pollution.

  • Have the English edited very carefully (subscript/superscript).

Have the English edited very carefully throughout the manuscript

  • Highlight key findings in the conclusions and compare them to other work. In general, expand the discussion in this regard.

The simultaneous generation of hot water and steam through the design of a pyroacuotubular (mixed) boiler contributes significantly to industry and society by optimizing the generation of hot water and steam and reducing the temperature of the gases emitted.

  • Remove less important and unrelated or less related references and ensure that all references are cited and arranged in the order required by the journal.

The authors evaluated the references and eliminated the less important, unrelated and less related references, citing and organizing them in the order required by the journal.

  • Avoid mass citations whenever possible, for example [2-5].

The  mass citations were avoided as can be verified in the manuscript

  1. “There is a growing development of technology dedicated to controlling pollutant gas emissions.” – There is no relevant mentioning, how this is mitigated nowadays, as well as, how emission treatment technologies are being developed, whether CO reduction or NOX decomposition.

Some related works are 10.1039/d0ra06969h, 10.1016/j.apcatb.2018.06.069, and 10.1016/j.renene.2017.09.063, but others can be considered as well. Presently, focus is only on upstream, while downstream (post-combustion, catalysis…) is important as well.

As suggestion references and text were added as follows in the manuscript

CO2 emissions also arise from some industrial processes as shown in Figure 1, and resource extraction, as well as from burning forests during land clearing. Some of these sources could supply decarbonized fuel, such as hydrogen, to the transportation, industrial, and construction sectors, thereby reducing emissions from these distributed sources.  Some industrial processes can also use and store small amounts of CO2 captured in manufactured products. CO2 is also emitted during certain industrial processes such as cement manufacturing or hydrogen production and during biomass combustion. Power plants and other large-scale industrial processes are prime candidates for CO2 capture and storage.

Figure 1. CO2 capture systems in industrial processes (adapted from BP).

There are several techniques to reduce CO2 pollution, among the most developed are post-combustion and oxy-combustion, pre-combustion and industrial processes from all compressing and dehydrating. For nox reduction, the CSR (Selective Catalytic Reduction) method can be used, whose efficiency is between 90 and 98 %, also combined with NSR (Nox Storage Reduction) catalysts.

Carbon Capture and Storage (CCS) is an alternative industry can have benefits from cutting industrial CO2 emissions deeply, and CCS is moving into carbon dioxide removal (CDR) in applications such as Direct Air Capture (DAC) and Bioenergy with CCS (BECCS), drawing down historical CO2 emissions from the atmosphere.

Oxyfuel Combustion (Figure 2) is another alternative to convert fossil fuel into carbon-neutral or carbon-negative with CO2 utilization/ storage by way of coping with the climate change issue, and will contribute to the environment and the economy of the community.

Figure 2. Oxyfuel combustion technology

  1. Figures should generally be referenced if they are borrowed from another work. If the Figures are the original, I think they should be of better quality.

The figures were referenced and their own were improved and others were added to improve understanding.

  1. Units should be consistent throughout the manuscript. For example, BHP and psi should be corrected in the manuscript. No interested reader will have to do the calculations himself.

The units were corrected throughout the manuscript, Table 2 and Table 3 were added to better explain that the SI units are taken as a reference, and the calculations and results were revised and corrected.

Table 2. Description of the SI units

Quantity

SI abbreviations

Force

N, mm

Mass

kg

Time

s, h

Stress, pressure

Pa, MPa

Energy, work, quantity of heat

J, kJ

Density

Kg/m3

Power

kW

Temperature

°C

Table 3. Description of the variables used in the calculations

Variable

Description

Do

Outside diameter

t

Plate thickness

L

Length of the ignition chamber in the water chamber zone

A

Calculation factor (Dimensionless)

R

Cylinder radius (variable)

t

Thickness of plate.

P

Boiler internal pressure

Pe

External pressure

B

Coefficient resulting from joining factor A and heating temperature.

e1,…e7

Plate thickness

S

Material yield stress, allowable material stress

E

Efficiency of the welded joint

P

Internal pressure supported by the vessel (0.758 MPa).

T

Dome Plate Thickness

D

Dome diameter

Mtotal

Water mass combustion chamber, water chamber, dome, metallic mass

CVF

Calorific value of fuel

Theoretical heat required

Combustion efficiency

Total heat

Flow Fuel mass

Heat required for phase change

Heat needed to heat the new volume of water in the dome

Heat required to heat water

Total heat required to heat water and generate steam

Internal volume of the steam chamber

Actual fuel mass

Combustion efficiency

Heat delivered to generate steam

Actual air-fuel ratio

Theoretical air-fuel ratio

%e

Percentage of excess air

  1. As mentioned in the general comments, there are many subscript and superscript errors in the text and equations that make the manuscript difficult to read. Correct the errors to make it easier to read and understand the proposed work.

The subscript and superscript errors in the text and equations were corrected throughout the manuscript, it was important for the comments to correct them, so Table 2 and Table 3 were added to better guide the calculations and their conversions.

  1. The conclusion lacks a detailed comparative discussion with references to similar studies using similar methods. Highlight key findings and advance the discussion by providing future perspectives.

At the reviewer's suggestion, the authors made the suggested changes.

Based on the research of [1] in which his new type of vertical mixed boiler with three passes or gas paths through its heat transfer surface for multipurpose use with alternative fuels optimizes thermal efficiency; by producing steam and hot water, contemplating the possibility of using alternative fuels such as diesel oil, natural gas and Liquefied Petroleum Gas (LPG), this new design reduces the emission of combustion gases since by introducing a volume of water in the combustion chamber surrounding the ignition chamber, energy is used, due to the fact that the combustion chamber contains an ignition chamber inside and its external surface will be surrounded by water that recirculates in order to take advantage of the combustion heat, obtaining hot water. The new proposal compared carbon emissions to the environment by fluid type; hot water, water steam, and parallel generation show that simultaneous generation generates greater benefits in relation to environmental pollution and the findings of this research lead us to conclude that the new design of the pyroacuotubular (mixed) boiler makes it possible to simultaneously generate hot water and water steam, and also to reduce the temperature of the gases emitted by increasing the thermal efficiency of the boiler, compared to other industrial boilers, with better use of the combustion gases, reducing the emission temperature of these gases, which in turn also reduces significatively the emission of greenhouse gas pollutants, contributing to reduce global warming and environmental pollution. On the other hand, previous studies of the modification made by [16] where was modifies a water-tube boiler are surpassed in efficiency by the present work,  as well as [17] the efficiency achieved in water-tube boilers shows that it is possible to modify to achieve a new design which is ratified and improved with the proposed design. Future prospects are encouraging considering that it is possible to improve the design by applying other methods and techniques based on thermodynamic laws.

Reviewer 2 Report

I more or less agree with the modifications, but it is necessary to correct the formal errors in the text:

It is not good when different unit systems are used. SI units should be used.

Alternatively, conversions to the SI system should be indicated for units of another type. Some values are listed in the SI system, but some are not.

-lines 52, 53, 54: it is not explained what "sCO2" is

-line 53: the subscript for "CO2" must be adjusted

-line 182: at least put the value "2psi" in "Pa" in parentheses

-line 249: it "1 atmosphere of pressure" technical or physical pressure, it is better to state in "Pa"

- Table 1:

- incorrect "Kw" correct is "kW"

- it would be appropriate to include the conversion of "lb/h" to the SI unit of the system

-line 343: incorrect "0,758" correct "0.758"

-line 352: typo "(Eq..2)"

-line 383: incorrect "3,33" correct "3.33"

-line 403 and 438: incorrect "0,758" correct "0.758"

-line 434: incorrect "6,25" correct "6.25"

I recommend checking every calculation where numbers are used and correcting the decimal point to a decimal dot.

-Table 5: in the header, adjust the units "Kg" to the correct form "kg", in Table 6 it is correct

-line734: incorrect "kj" correct "kJ"

For calculations and various mathematical relationships, it is advisable to add the Nomenclature to the manuscript, which I already wrote about in the first review.

Author Response

Response reviewer comments

I more or less agree with the modifications, but it is necessary to correct the formal errors in the text:

We would like to start by thanking the reviewer for this valuable feedback. We acknowledge the importance of the reviewer's questions and recommendations and would like to address them point-by-point.

It is not good when different unit systems are used. SI units should be used.

Thanks for the valuable comment. We agree with this comment, and we have made some changes by giving more details about the SI units that we will show with the calculations. The following Table 2 has been added in the related work section, page 8 of the manuscript.

Table 2. Description of the SI units

Quantity

SI abbreviations

Force

N, mm

Mass

kg

Time

s, h

Stress, pressure

Pa, MPa

Energy, work, quantity of heat

J, kJ

Density

Kg/m3

Power

kW

Temperature

°C

Alternatively, conversions to the SI system should be indicated for units of another type. Some values are listed in the SI system, but some are not.

The authors have taken the above recommendation as a guide and the conversions have been made according to Table 2 below.

-lines 52, 53, 54: it is not explained what "sCO2" is

Changes required to lines 52,53 and 54 were performed and the meaning of sCO2 was explained in the include following paragraph

The behavior of carbon dioxide is as a gas in air at standard temperature and pressure or as a solid (dry ice), if the temperature and pressure are increased from the standard point to the critical point of carbon dioxide it can adopt properties between a gas and a liquid and behave as a supercritical fluid above its critical temperature, in this way supercritical carbon dioxide (sCO2) maintains its critical or above standard temperature and pressure. Supercritical carbon dioxide (sCO2) uses CO2 as the working fluid in small turbomachines. Power cycles based on sCO2 have the potential for increased heat-to-electricity conversion efficiency, high power density, and simplicity of operation compared to existing steam-based power cycles.

-line 53: the subscript for "CO2" must be adjusted

Line 53 was adjusted in the subscript for "CO2".

-line 182: at least put the value "2psi" in "Pa" in parentheses

Line 182 was adjusted with the value in Pa.

-line 249: it "1 atmosphere of pressure" technical or physical pressure, it is better to state in "Pa"

Line 249: was adjusted with the value in Pa

- Table 1:- incorrect "Kw" correct is "kW"

Table 1 was corrected with "kW"

Table 1.  Technical specifications

Design

Pyroacuotubular (mixed)  

Power

15 BHP = 150 kW

Transfer area

75 feet2 = 6.96 m2

Theoretical steam flow

517.5 lb/h = 234.73 kg/h

Design pressure

110 psig = 0.758 MPa

Maximum working pressure

100 psig = 0.689 MPa

Position

Vertical

Number of gas passages

03

Fuels to be used

Liquefied petroleum gas (LPG)

Simultaneous thermal fluids

Steam and hot water

- it would be appropriate to include the conversion of "lb/h" to the SI unit of the system

As shown in Table 1 was corrected adding the conversion in "kg/h"

-line 343: incorrect "0,758" correct "0.758"

Line 343 was corrected with "0.758"

-line 352: typo "(Eq..2)"

Line 352 was corrected with “The vessels subjected to external pressure (P) is calculated in Equation (2)”

-line 383: incorrect "3,33" correct "3.33"

Line 383 was corrected with "3.33"

-line 403 and 438: incorrect "0,758" correct "0.758"

Line 403 and 438 were corrected with "0.758"

-line 434: incorrect "6,25" correct "6.25"

Line 434 was corrected with "6.25"

I recommend checking every calculation where numbers are used and correcting the decimal point to a decimal dot.

The authors have reviewed all calculations using numbers and have corrected and standardized the units according to SI units, which is why we have added Table 2 as a guide and reference for correcting and converting our calculations.

-Table 5: in the header, adjust the units "Kg" to the correct form "kg", in Table 6 it is correct

Table 5, now Table 7 was corrected and added the units in kg.

Table 7. Amount of combustion gases emitted to the environment

Fluid type

Emission

Combustion gases emitted into the environment

kg /h

kg /day

kg /month

kg /year

% Reduction

Hot water

CO2

992.00

23 808.00

714 240.00

8 689 920.00

44.35

CO

3.05

73.30

2,198.88

26 753.04

44.27

Steam

CO2

790.72

18,977.28

569 318.40

6 926 707.20

55.65

CO

2.43

58.32

1 749.60

21,286.80

55.66

Parallel generation

CO2

1 782.72

42 785.28

1 283 558.40

15 616 627.20

0.00

CO

5.48

131.52

3 945.60

48 004.80

0.00

Simultaneous generation

CO2

790.72

18 977.28

569 318.40

6 926 707.20

55,65

CO

2.43

58.32

1 749.60

21 286.80

55.66

-line734: incorrect "kj" correct "kJ"

Line 734 was corrected with "kJ"

For calculations and various mathematical relationships, it is advisable to add the Nomenclature to the manuscript, which I already wrote about in the first review.

Following your recommendations, for calculations and various mathematical relationships and nomenclature, Table 2 and Table 3 have been added for guidance and reference, as well as revised calculations and resulting values.

Table 2. Description of the SI units

Quantity

SI abbreviations

Force

N, mm

Mass

kg

Time

s, h

Stress, pressure

Pa, MPa

Energy, work, quantity of heat

J, kJ

Density

Kg/m3

Power

kW

Temperature

°C

Table 3. Description of the variables used in the calculations

Variable

Description

Do

Outside diameter

t

Plate thickness

L

Length of the ignition chamber in the water chamber zone

A

Calculation factor (Dimensionless)

R

Cylinder radius (variable)

t

Thickness of plate.

P

Boiler internal pressure

Pe

External pressure

B

Coefficient resulting from joining factor A and heating temperature.

e1,…e7

Plate thickness

S

Material yield stress, allowable material stress

E

Efficiency of the welded joint

P

Internal pressure supported by the vessel (0.758 MPa).

T

Dome Plate Thickness

D

Dome diameter

Mtotal

Water mass combustion chamber, water chamber, dome, metallic mass

CVF

Calorific value of fuel

Theoretical heat required

Combustion efficiency

Total heat

Flow Fuel mass

Heat required for phase change

Heat needed to heat the new volume of water in the dome

Heat required to heat water

Total heat required to heat water and generate steam

Internal volume of the steam chamber

Actual fuel mass

Combustion efficiency

Heat delivered to generate steam

Actual air-fuel ratio

Theoretical air-fuel ratio

%e

Percentage of excess air

Round 3

Reviewer 2 Report

On the professional side, I have no comments. I agree with the modifications, but it is necessary to modify a few more formal errors in the text:

-line 57, 93: the subscript for "CO2" must be adjusted

-line 314: It is "1 atmosphere of pressure" technical (98,1 kPa) or physical (101,325 kPa) pressure, it is better to state in the SI system of the unit

-Table 2: Density unit is incorrectly written "Kg/..." -> "kg/..."

-line 420: incorrect "0,758" correct "0.758"

-line 878: the unit is missing, according to the calculations I assume it is "kg/kmol"

-line 1057: incorrect "C02" correct "CO2"

- line 741, ...: it is not standard for numbers greater than a million to use a separator like "1'023 984.21" when spaces have already been used between the numbers. It would be better to use a space instead of the specified separator. In Table 8, this is no longer the case and it is a common standard. Please unify the writing of large numbers.